# MM-Eureka: Toward Stable Multimodal Reasoning via Rule-based Reinforcement Learning with Policy Drift Control

**Fanqing Meng**[1,*] **Lingxiao Du**[1,*] **Zongkai Liu**[2,*] **Zhixiang Zhou**[2,*]

**Quanfeng Lu**[3] **Tiancheng Han**[2] **Daocheng Fu**[4] **Kaipeng Zhang**[4]

**Ping Luo**[3] **Yu Qiao**[4] **Jiaheng Zhang**[1]

**Michael Qizhe Shieh**[1,†] **Qiaosheng Zhang**[4,†] **Wenqi Shao**[4,†]

[1]**National University of Singapore** [2]**Shanghai Innovation Institute**
[3]**The University of Hong Kong** [4]**Shanghai AI Laboratory**

[*]Equal contribution [†]Corresponding author

**Reviewed on OpenReview:** `https://openreview.net/forum?id=8y1ch6y24H`

## Abstract

Existing rule-based reinforcement learning (RL) methods that work well for text reasoning often collapse when extended to long-horizon multimodal reasoning settings. We identify a structural instability driven by ratio-based policy objectives under sparse multimodal rewards: importance sampling ratios in PPO-style objectives can amplify policy shifts, especially under negative advantages, which can trigger catastrophic mid-training collapse. To make multimodal rule-based RL reliably trainable, we propose **CPGD (Clipped Policy Gradient Optimization with Policy Drift)**, a stability-oriented RL objective that removes ratio-induced amplification while maintaining proximal updates via an explicit policy drift regularizer and a numerically stable KL estimator. We provide both theoretical analysis and empirical evidence showing that ratio-based objectives can systematically amplify policy drift beyond intended bounds under sparse-reward multimodal settings, and demonstrate how CPGD addresses this through controlled policy updates. To support diagnosis and evaluation under consistent settings, we introduce **MMK12**, a K12-level multimodal reasoning dataset with 15,616 training problems and 2,000 evaluation questions across mathematics, physics, chemistry, and biology, all with human-verified solutions. Using CPGD on MMK12, we train **MM-Eureka** models that demonstrate stable long-horizon training without collapse. CPGD achieves consistent performance improvements while maintaining training stability throughout, validating that the instability mechanism has been effectively addressed. We open-source our complete pipeline at `https://github.com/ModalMinds/MM-EUREKA`

## 1 Introduction

Large-scale reinforcement learning (RL) (Sutton et al., 1998) has demonstrated remarkable progress in improving the reasoning ability of Large Language Models (LLMs), particularly in math and code domains (OpenAI, 2024; DeepSeek-AI et al., 2025). Recent work such as o1 (OpenAI, 2024) and DeepSeek-R1 (DeepSeek-AI et al., 2025) shows that rule-based RL can achieve breakthrough improvements in complex reasoning tasks. However, many real-world reasoning problems—such as interpreting scientific diagrams, analyzing geometric figures, or solving visual math problems—fundamentally require multimodal understanding. Despite the success of rule-based RL in text-only settings, extending these methods to long-horizon multimodal reasoning remains challenging.

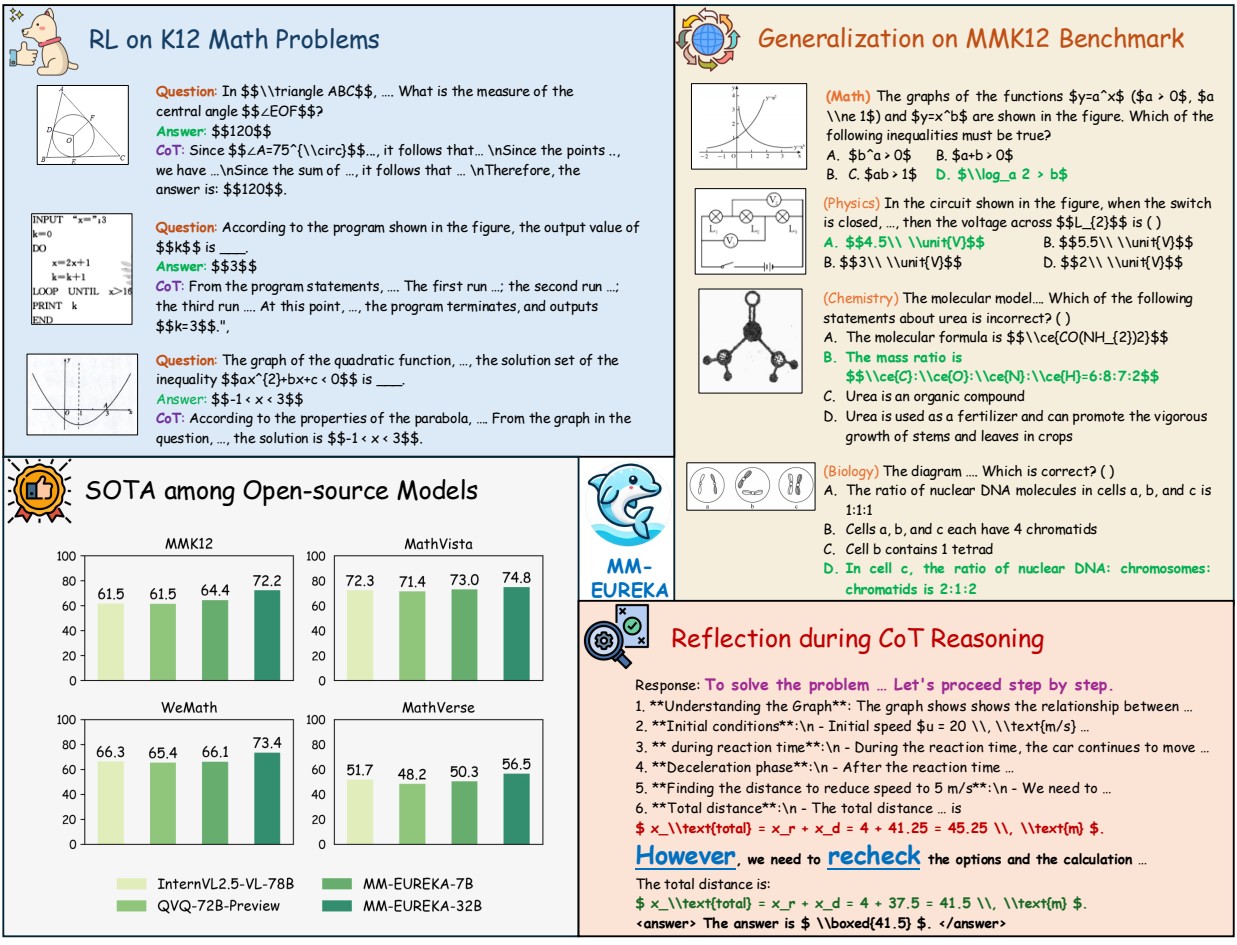

Figure 1: Overview of MMK12 and MM-Eureka. MMK12 training set contains diverse multimodal mathematical questions with verified answers and solution processes, while its evaluation set contains multiple-choice questions across mathematics, physics, chemistry, and biology. MM-Eureka, trained with CPGD on MMK12, achieves strong multimodal reasoning performance and exhibits aha-moment behaviors similar to DeepSeek-R1.

A recurring issue in multimodal rule-based RL is catastrophic training collapse. Recent attempts to transfer rule-based RL techniques from text to multimodal domains (Chen et al., 2025; Peng et al., 2025; Team et al., 2025a) have encountered a critical obstacle: training often *catastrophically collapses* mid-training. Through extensive experiments with existing RL algorithms such as GRPO (DeepSeek-AI et al., 2025), RLOO (Kool et al., 2019; Ahmadian et al., 2024), and REINFORCE++ (Hu, 2025), we observe that models frequently experience sudden failures where accuracy rewards drop to near-zero and outputs degenerate into trivial patterns (e.g., emitting only format tokens without meaningful reasoning). This instability makes long-horizon training—the key ingredient enabling reasoning breakthroughs in text-only settings—infeasible in multimodal scenarios.

We trace this instability to a structural mechanism in ratio-based policy objectives. Existing methods incorporate importance sampling ratios $\pi_\theta/\pi_{\text{old}}$ directly in their gradient-carrying loss terms. While PPO-clip (Schulman et al., 2017) is commonly used to constrain extreme policy updates, we show that its one-sided clipping mechanism *fails to prevent ratio amplification*, especially under negative advantages. This allows the policy ratio to grow unconstrained in harmful directions, causing policy drift to exceed intended bounds. In multimodal settings with large action spaces and sparse rewards, this amplification effect is particularly severe: a few poor samples with large ratios can dominate gradients and trigger collapse.

To address this failure mode, we propose **CPGD (Clipped Policy Gradient Optimization with Policy Drift)**, a stability-oriented RL objective that directly addresses ratio-induced amplification. The key insight is to *remove ratios from the gradient-carrying term* while maintaining proximal updates through an explicit policy drift regularizer. Specifically, CPGD replaces the PPO-clip loss with a REINFORCE-style loss (Sutton et al., 1998), applies clipping on the *logarithm* of the ratio (which prevents ratio explosion), and introduces a forward KL-based drift penalty with a numerically stable estimator. We provide both theoretical analysis (Proposition 1 and Theorem 1) and empirical evidence showing that this design prevents the amplification effect while preserving monotonic improvement guarantees.

To systematically diagnose collapse and evaluate stability under controlled settings, we introduce **MMK12**, a K12-level multimodal reasoning dataset with 15,616 training problems and 2,000 evaluation questions spanning mathematics, physics, chemistry, and biology, all with human-verified solutions. Using CPGD on MMK12, we train **MM-Eureka** models (7B and 32B variants) that demonstrate stable long-horizon training without collapse. On Qwen2.5-VL-7B, CPGD achieves 10% overall improvement, while MM-Eureka-32B reaches competitive performance compared to much larger models. Critically, training remains stable throughout—no mid-training collapse, no degenerate outputs—validating that the instability mechanism has been effectively addressed.

Through our work, we observe several insights about multimodal RL: **(1)** RL training increases the probability of applying existing knowledge correctly rather than injecting new knowledge; **(2)** training on mathematics leads to improvements in physics, chemistry, and biology; **(3)** RL generalizes better than supervised methods (SFT (Ouyang et al., 2022), COT SFT (Guo et al., 2024)) across diverse reasoning tasks.

We summarize our contributions as follows:

- **A stability-oriented RL objective (CPGD)** that removes ratio-induced amplification. We introduce CPGD, a drift-controlled policy gradient framework that avoids placing importance ratios in the gradient-carrying term, maintains proximal updates via explicit policy drift regularization, and uses a numerically stable KL estimator to prevent runaway updates.

- **An enabling dataset/benchmark (MMK12)** used to observe and diagnose instability. We introduce MMK12, a K12-level multimodal reasoning dataset with 15,616 training problems and 2,000 evaluation questions (mathematics, physics, chemistry, biology), designed to support reliable rule-based training and make the failure mode reproducible under consistent settings.

- **Demonstrated stable training with MM-Eureka models.** Using CPGD on MMK12, we train MM-Eureka-7B and MM-Eureka-32B that maintain stability throughout long-horizon training. On Qwen2.5-VL-7B, CPGD achieves 10% overall improvement without collapse, validating that the instability mechanism has been addressed.

- **Open-source pipeline for reproducibility.** We release the complete pipeline including MMK12 dataset, CPGD implementation, trained models, and training recipes to enable community verification and extension of our findings.

## 2 Related Work

### 2.1 Language Reasoning Model

LLMs have demonstrated impressive performance across a wide range of tasks, yet more complex challenges require these models to exhibit human-like reasoning capabilities. As a result, enhancing the reasoning ability of LLMs has become a critical research focus. Reinforcement Learning from Human Feedback (RLHF), particularly Proximal Policy Optimization (PPO) (Schulman et al., 2017), has shown promise in enabling LLMs to learn reasoning abilities effectively. However, the PPO training process is computationally intensive and complex, prompting the development of simplified alternatives such as Direct Preference Optimization (DPO) (Rafailov et al., 2023). While DPO alleviates some training difficulties, its reliance on offline data can limit model performance. To address these limitations, methods like Group Relative Policy Optimization

(GRPO) (DeepSeek-AI et al., 2025), REINFORCE Leave-One-Out (RLOO) (Kool et al., 2019; Ahmadian et al., 2024), and Reinforce++ (Hu, 2025) have been introduced. Notably, Deepseek R1 (DeepSeek-AI et al., 2025) reveals that pure RL can encourage LLMs to actively engage in reasoning, including self-reflection and error correction. Despite these advancements, research on improving the reasoning capabilities of multimodal large models remains relatively scarce, highlighting an important direction for future exploration.

### 2.2 Vision-Language Reasoning Model

Currently, the leading models in multimodal reasoning are closed-source systems such as GPT-4o (Hurst et al., 2024) and Kimi-VL (Team et al., 2025b). In contrast, the open-source community remains noticeably behind, still in the early stages of exploration. Recent concurrent efforts have begun to explore the use of RL to enhance the visual reasoning capabilities of vision-language reasoning models (VLMs), aiming to trigger an "Aha Moment" in visual reasoning. LMM-R1 (Peng et al., 2025) strengthens visual reasoning through a two-stage rule-based RL approach; however, its primary reasoning performance gains are derived from text-only datasets rather than genuinely multimodal datasets. R1-V (Chen et al., 2025) investigates rule-based RL within specific subdomains, such as geometric reasoning and object counting tasks, but falls short of addressing more complex reasoning challenges. Reason-RFT (Tan et al., 2025), on the other hand, relies on SFT with COT reasoning activation data to achieve an effective cold start before the RL training phase. In this paper, our objective is to develop an effective, stable, and comprehensive open-source training pipeline for multimodal reasoning models, including datasets, code, and models. Our work aims to advance the growth and innovation of the open-source community.

## 3 MMK12: A Multimodal Mathematic K12-Level Dataset

|  | Scope | Type | Img. Source | QA Source | CoT Answer Source |
|---|---|---|---|---|---|
| MAVIS (Zhang et al., 2024b) | Geo & Func | MCQ & FB | Synthetic | Synthetic Engine | GPT-4o |
| Geo3k (Lu et al., 2021) | Geo | FB | Real world | Real world | None |
| RCOT (Deng et al., 2024) | Geo | MCQ & FB | Synthetic | Synthetic Engine | GPT-4o |
| MultiMath (Peng et al., 2024) | Diverse | MCQ & FB | Real World | GPT-4o | GPT-4o |
| MMK12 | Diverse | FB | Real World | Real World | Real World |

Table 1: Comparison of dataset characteristics with other multimodal mathematical reasoning datasets. MMK12 comprises more diverse and high-quality questions, with guaranteed correct answers and solution processes. Abbreviations: Geo = Geometry, Func = Function, MCQ = Multiple-Choice Question, FB = Fill-in-the-Blank, Img. = Image, QA = Question-Answer, CoT = Chain-of-Thought.

As shown in Table 1, current multimodal mathematical reasoning datasets have limited scope and face challenges in ensuring answer correctness. For instance, while RCOT and MAVIS maintain answer accuracy through synthetic engine-generated QA pairs, this approach restricts problem diversity. Geo3k manually collected 3,000 geometry problems with verified answers, but it focuses solely on geometry examples. Although MultiMath gathers problems from real-world scenarios to ensure diversity, its reference answers generated by GPT-4o cannot guarantee correctness.

To address these limitations, we introduce MMK12, a new dataset comprising over 15,000 multimodal mathematical reasoning problems across a wide range of domains, including geometry, functions, and graphical reasoning. Each problem is accompanied by a standard reference answer and a detailed step-by-step solution to ensure both accuracy and interpretability.

As illustrated in Figure 2, MMK12 is collected from Chinese textbooks and examination papers covering elementary to high school levels, then translated and verified using LLMs. The training set comprises 15,616 multimodal fill-in-the-blank mathematics problems (455 elementary, 9,776 middle school, 5,385 high school), with each sample including the question, image, final answer, and CoT-formatted solution. The evaluation set contains 2,000 multiple-choice questions across math, physics, chemistry, and biology (500 per discipline). All problems span diverse knowledge domains including function reasoning, geometric reasoning, and pattern reasoning. Detailed construction procedures and problem categories are described in Appendix B.

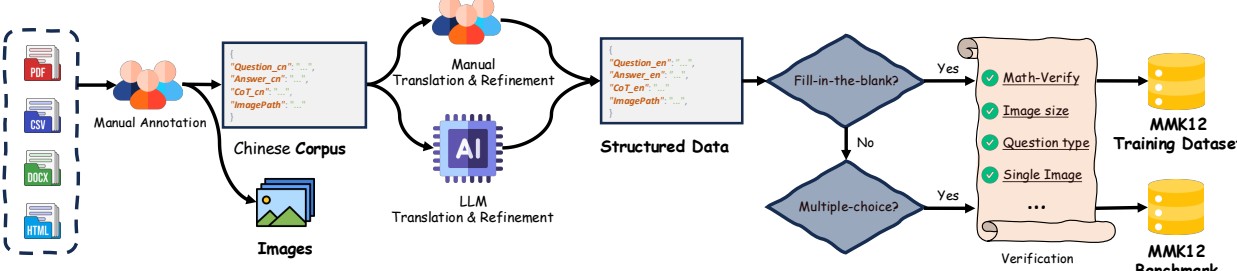

Figure 2: The construction overview of MMK12. We collect diverse K12-level multimodal math problems from multiple sources, convert them to standardized English using LLMs, and verify all content for accuracy. The resulting MMK12 dataset includes a training set of 15,616 samples and a test set with 500 multiple-choice questions each for math, physics, chemistry, and biology.

MMK12's standardized construction with human verification ensures both diversity and correctness, making it suitable for RL and SFT training while providing reliable multidisciplinary reasoning evaluation. Examples are shown in Figure 1.

## 4 Method

This section introduces *Clipped Policy Gradient Optimization with Policy Drift* (CPGD), our stability-oriented RL objective designed to make multimodal rule-based RL reliably trainable.

Existing ratio-based objectives place importance sampling ratios $\pi_\theta/\pi_{\mathrm{old}}$ directly in the gradient-carrying loss term, which we identify as the source of training instability. While these ratios correct for distribution mismatch, they simultaneously amplify policy drift: even with PPO-clip constraints, ratios can grow beyond intended bounds (especially under negative advantages), causing a few poor samples to dominate gradients and trigger collapse. CPGD addresses this by *removing ratios from the gradient term* and instead using an explicit policy drift regularizer to maintain proximal updates. This structural change eliminates ratio-induced amplification while preserving the benefits of controlled policy optimization.

In Section 4.2, we present the CPGD objective with theoretical analysis showing why ratio-based objectives can systematically amplify policy drift in sparse-reward settings and how CPGD prevents this. Section 4.3 provides empirical evidence of collapse in existing methods and demonstrates CPGD's stability. Section 4.4 describes the practical implementation, including a numerically stable KL estimator that balances theoretical correctness with empirical robustness.

### 4.1 Basic settings

We use Qwen2.5-VL (Bai et al., 2023) with 7B and 32B parameters as our base models. Our reinforcement learning (RL) approach follows a similar design to DeepSeek-R1 (DeepSeek-AI et al., 2025), we also adopt the simple rule-based reward function rather than using outcome or process reward models, thereby alleviating reward hacking (Gao et al., 2022). Specifically, we employ rule-based rewards for output formatting ($r_{\mathrm{format}} \in \{0, 0.5\}$) and accuracy ($r_{\mathrm{accuracy}} \in \{0, 1\}$). More detail in Appendix C.1.

### 4.2 Clipped Policy Gradient Optimization with Policy Drift (CPGD)

Under the response-level MDP assumption, CPGD aims to maximize the following formula:

$$\mathcal{L}_{\mathrm{CPGD}}(\theta; \theta_{old}) = \mathbb{E}_{\mathbf{x} \sim \mathcal{D}} \Big[ \mathbb{E}_{\mathbf{y} \sim \pi_{\theta_{old}}(\cdot|\mathbf{x})} \big[ \Phi_\theta(\mathbf{x}, \mathbf{y}) \big] - \alpha \cdot D_{\mathrm{KL}}(\pi_{\theta_{old}}, \pi_\theta|\mathbf{x}) \Big], \quad (1)$$

where

$$\Phi_\theta(\mathbf{x}, \mathbf{y}) := \min\left(\ln\frac{\pi_\theta(\mathbf{y}|\mathbf{x})}{\pi_{\theta_{old}}(\mathbf{y}|\mathbf{x})} \cdot A^{\mathrm{CPGD}}(\mathbf{x}, \mathbf{y}), \mathrm{clip}_{\ln(1-\epsilon)}^{\ln(1+\epsilon)}\left(\ln\frac{\pi_\theta(\mathbf{y}|\mathbf{x})}{\pi_{\theta_{old}}(\mathbf{y}|\mathbf{x})}\right) A^{\mathrm{CPGD}}(\mathbf{x}, \mathbf{y})\right),$$

$$A^{\mathrm{CPGD}}(\mathbf{x}, \mathbf{y}) := \mathcal{R}_o(\mathbf{x}, \mathbf{y}) - \mathbb{E}_{\mathbf{y}' \sim \pi_\theta(\cdot|\mathbf{x})}\left[\mathcal{R}_o(\mathbf{x}, \mathbf{y}')\right],$$

$$D_{\mathrm{KL}}(\pi_{\tilde\theta}, \pi_\theta|\mathbf{x}) := \mathbb{E}_{\mathbf{y} \sim \pi_{\tilde\theta}(\cdot|\mathbf{x})}\left[\ln\frac{\pi_{\tilde\theta}(\mathbf{y}|\mathbf{x})}{\pi_\theta(\mathbf{y}|\mathbf{x})}\right].$$

Hereinafter, we term the KL divergence between the old and current policies as *policy drift*, and between the current and reference policies as *reference constraint*. CPGD differs from the standard PPO-clip loss in two key aspects: (1) REINFORCE loss ($\ln\frac{\pi_\theta(\mathbf{y}|\mathbf{x})}{\pi_{\theta_{old}}(\mathbf{y}|\mathbf{x})}$) with the PPO-clip's clip mechanism is used. (2) A PPO-KL like policy drift is introduced, imposing a forward KL divergence penalty between the old and current policies $D_{\mathrm{KL}}(\pi_{\theta_{old}}, \pi_\theta|\mathbf{x})$.

CPGD adopts a REINFORCE-style gradient formulation (logarithm of ratio multiplied by advantage) to decouple policy optimization from direct ratio amplification. In the original PPO objective, the importance-sampling ratio corrects for distribution mismatch between old and current policies, but simultaneously introduces high variance that can destabilize training. As empirically demonstrated in Section 4.3, this variance can cause catastrophic training collapse, while using a REINFORCE loss without the ratio substantially improves training stability. Proposition 1 provides theoretical justification: the use of policy ratios in gradient-carrying terms amplifies policy drift, causing the updated policy to exceed intended bounds even under PPO-clip constraints.

Policy drift regularization and clipping are introduced to enforce proximal policy updates while avoiding ratio-induced amplification. These mechanisms are critical for the monotonic improvement guarantees established in Theorem 1 and for mitigating reward hacking behaviors such as length collapse (see Section 4.3). The clip mechanism reduces the need for a large weight on the policy drift term: when the policy stays within the clipping range, the drift term remains small, allowing the algorithm to focus on optimizing the main objective $\Phi$. If the policy strays beyond the range, the main objective's gradient is clipped to zero, and the drift term takes over to correct the deviation.

The following proposition formalizes why ratio-based clipping alone cannot prevent policy drift amplification, even under small learning rates.

**Proposition 1.** *Let $\theta_0$ be a parameter such that the importance-sampling ratio satisfies $|\frac{\pi_{\theta_0}(\mathbf{y}|\mathbf{x})}{\pi_{\theta_{old}}(\mathbf{y}|\mathbf{x})} - 1| = \epsilon$. Consider updating $\theta_0$ using either (i) the PPO-clip objective, resulting in parameter $\theta_1^{PPO}$, or (ii) the CPGD objective with $\alpha = 0$ (denoted as CPG), yielding parameter $\theta_1^{CPG}$. Then, there exists a constant $\eta_{\max} > 0$ such that for any learning rate $\eta \in (0, \eta_{\max})$, the following inequality holds:*

$$\left|\frac{\pi_{\theta_1^{PPO}}(\mathbf{y}|\mathbf{x})}{\pi_{\theta_{old}}(\mathbf{y}|\mathbf{x})} - 1\right| > \left|\frac{\pi_{\theta_1^{CPG}}(\mathbf{y}|\mathbf{x})}{\pi_{\theta_{old}}(\mathbf{y}|\mathbf{x})} - 1\right| > \left|\frac{\pi_{\theta_0}(\mathbf{y}|\mathbf{x})}{\pi_{\theta_{old}}(\mathbf{y}|\mathbf{x})} - 1\right| = \epsilon.$$

*After one update step, both PPO and CPG increase the importance-sampling ratio deviation from the old policy, but PPO does so more aggressively than CPG.*

The following theorem further presents that CPGD enjoys the monotonic improvement guarantee, indicating its theoretical rationality. See Appendix A for the proofs of Proposition 1 and Theorem 1.

**Theorem 1.** *Let $\{\pi_{\theta_k}\}_{k=0}^\infty$ denote the sequence of policies generated by the CPGD update rule: $\theta_{k+1} = \arg\max_\theta \mathcal{L}_{CPGD}(\theta; \theta_{old})$ where the advantage function is competed as $A^{CPGD}(\mathbf{x}, \mathbf{y}) = \mathcal{R}_o(\mathbf{x}, \mathbf{y})$. Then, $\pi_{\theta_{k+1}}$ is better than $\pi_{\theta_k}$, i.e., $\eta(\theta_{k+1}) \geq \eta(\theta_k)$, where $\eta(\theta) := \mathbb{E}_{\mathbf{x}\sim\mathcal{D}, \mathbf{y}\sim\pi_{\theta_{old}}(\cdot|\mathbf{x})}[\mathcal{R}_o(\mathbf{x}, \mathbf{y})].$*

## 4.3 Training collapse

Several studies suggest that the reference constraint may hinder policy improvement (Yu et al., 2025; Hu et al., 2025). However, we observe that removing this KL term leaves the PPO-clip loss alone insufficient to effectively constrain large policy shifts, which can lead to training collapse. While such collapse may be

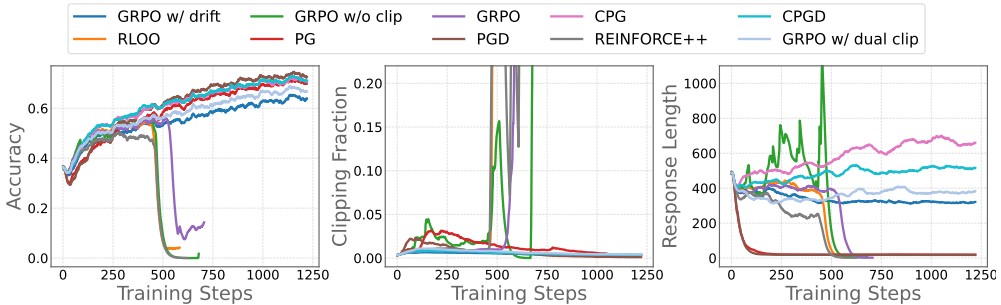

Figure 3: Accuracy, clipping fraction and response length curves throughout training.

partially mitigated through techniques such as early stopping or small learning rates, it remains a latent instability that undermines the reliability of continued training. In this subsection, we examine training collapse and show that CPGD effectively prevents it.

Figure 3 presents training curves on the MMK12 dataset for RLOO, REINFORCE++, GRPO, GRPO w/o clip (i.e., GRPO without the clip mechanism), GRPO w/ dual clip (i.e., the policy ratio is additionally clipped to no more than a constant—3.0 in our case—when advantage is negative (Ye et al., 2020)), GRPO w/ drift (i.e., GRPO with policy drift), PG (basic policy gradient), CPG (PG with the clip mechanism), PGD (PG with the policy drift), and CPGD, all without the reference constraint. We use Qwen2.5-VL-7B (Bai et al., 2023) as the base model. All algorithms share the same hyperparameters: a training and rollout batch size of 128, 8 responses per prompt, a learning rate of $1e-6$, one PPO epoch, and ten training episodes.

As shown in Figure 3, methods such as REINFORCE++, RLOO, GRPO w/o clip, and GRPO exhibit highly unstable policy ratio dynamics, leading to training collapse in mid stages. In contrast, GRPO w/ dual clip, GRPO w/ drift, PG, CPG, PGD, and CPGD maintain stable training curves. GRPO w/ dual clip mitigates instability by globally constraining the policy ratio, while the PG series sidesteps ratio-induced variance by excluding it from the loss computation. These comparisons indicate that incorporating policy ratios in the loss can introduce high variance during fluctuations, and that simple one-sided clipping fails to recover from extreme ratios, ultimately causing collapse. Although dual clip mechanism stabilizes training, it may introduce new issues: frequent zero-gradient updates and ineffective learning under negative advantages due to the zero-gradient clipped large ratios. Additionally, GRPO w/ drift demonstrates that incorporating policy drift effectively constrains the policy ratio within a reasonable range, thereby preventing training collapse.

We further verify that these findings generalize beyond MMK12 and are robust to hyperparameter choices. Additional collapse studies on the DeepVision dataset (Sun et al., 2026) and sensitivity analyses across different batch sizes and learning rates are provided in Appendix E.

On the other hand, while prior work suggests clipping may be unnecessary due to the low proportion of clipped ratios (Ahmadian et al., 2024), our findings suggest otherwise. Despite only ∼1% of ratios being clipped, training performance diverges significantly with and without clipping. Specifically, methods like PG and PGD—though stable without ratio terms—suffer from response length collapse, degenerating into trivial outputs (e.g., only emitting tokens like <think>) that exploit the format reward function without performing meaningful reasoning. This highlights the model's vulnerability to reward hacking, likely due to overly aggressive updates. These results reveal the necessity of the proximal policy updates.

## 4.4 Implementation

In this subsection, we design a practically implementable loss in per-token form based on the CPGD update formulation (Equation 1), aiming to strike a balance between theoretical rigor and empirical applicability. This practical loss is straightforward to integrate into widely-used LLM training frameworks like Open-

RLHF (Hu et al., 2024) and veRL (Sheng et al., 2024):

$$\mathcal{J}_{\text{CPGD}}(\theta) = -\frac{1}{|\mathcal{D}|} \sum_{(\mathbf{x}, \{\mathbf{y}^{(k)}\}_{k=1}^K) \in \mathcal{D}} \frac{1}{\sum_{k=1}^K |\mathbf{y}^{(k)}|} \left[ \sum_{i=1}^{|\mathbf{y}^{(k)}|} \left( \Phi_\theta^i(\mathbf{x}, \mathbf{y}^{(k)}) - \alpha \cdot \mathcal{E}_{\theta_{old},\theta}^i(\mathbf{x}, \mathbf{y}^{(k)}) \right) \right], \quad (2)$$

where

$$\Phi_\theta^i(\mathbf{x}, \mathbf{y}) := \min\left( \ln \frac{\pi_\theta(y_i|\mathbf{x}, \mathbf{y}_{<i})}{\pi_{\theta_{old}}(y_i|\mathbf{x}, \mathbf{y}_{<i})} \cdot A_\omega^{\text{CPGD}}(\mathbf{x}, \mathbf{y}), \text{clip}_{\ln(1-\epsilon_i)}^{\ln(1+\epsilon_i)} \left( \ln \frac{\pi_\theta(y_i|\mathbf{x}, \mathbf{y}_{<i})}{\pi_{\theta_{old}}(y_i|\mathbf{x}, \mathbf{y}_{<i})} \right) A_\omega^{\text{CPGD}}(\mathbf{x}, \mathbf{y}) \right),$$

$$A_\omega^{\text{CPGD}}(\mathbf{x}, \mathbf{y}^{(k)}) := \omega(\mathbf{x}) \cdot \left( \mathcal{R}_o(\mathbf{x}, \mathbf{y}^{(k)}) - \text{mean}\left( \{\mathcal{R}_o(\mathbf{x}, \mathbf{y}^{(k')})\}_{k'=1}^K \right) \right),$$

$$\mathcal{E}_{\theta_{old},\theta}^i(\mathbf{x}, \mathbf{y}) := \min\left( \frac{\text{sg}(\pi_\theta(y_i|\mathbf{x}, \mathbf{y}_{<i}))}{\pi_{\theta_{old}}(y_i|\mathbf{x}, \mathbf{y}_{<i})} - 1, c \right) \cdot \ln \pi_\theta(y_i|\mathbf{x}, \mathbf{y}_{<i}).$$

Here, $\text{sg}(\cdot)$ denotes the operation that prevents gradient computation, $\omega(\mathbf{x})$ is a per-prompt weighting factor, and $c > 0$ is a constant. Each modification from the theoretical formulation (Equation 1) to the practical loss (Equation 2) is a principled adaptation rather than an ad-hoc choice; we provide ablation studies isolating each component's contribution in Appendix C.8. We provide the following clarifications regarding the differences:

**(I) Policy optimization term:** In the theoretical update (Equation 1), the policy optimization term is written in the form of joint distribution. However, in the practical implementation (Equation 2), it is decomposed into token level using the decomposability of the logarithm function. Specifically, the clipping threshold $\epsilon_i$ can be set the same for all tokens, ensuring that each token shares the same clip range. Alternatively, a tight-to-loose schedule can be employed such as $\epsilon_i = \lambda\epsilon + (1 - \lambda)\epsilon \cdot i/|\mathbf{y}^{(k)}|$, which assigns smaller thresholds to earlier tokens that usually have higher variance.

**(II) Policy drift:** Policy drift leverages the decomposability of the logarithm function and requires careful estimation. Standard KL estimators ($k_1$ and $k_3$) have limitations: $k_1$ provides incorrect gradient direction while $k_3$ suffers from numerical instability when policy ratios are large. To address this, we propose a novel clipped gradient estimator $\mathcal{E}_{\theta_{old},\theta}^i$ that combines correct corrective direction with numerical stability. Detailed derivations and comparisons of different KL estimators are provided in Appendix C.9.

**(III) Weighted advantage:** In the view of the response level, each prompt can be viewed as a distinct task. Consequently, we can introduce a per-prompt weighting factor $\omega(\mathbf{x})$ to assign different levels of importance to different prompts. (1) *Equal weight*: when $\omega(\mathbf{x}) = 1$, $A_\omega^{\text{CPGD}}$ reduces to the original unweighted form. (2) *STD weight*: when $\omega(\mathbf{x}) = 1/\text{std}(\{\mathcal{R}(\mathbf{x}, \mathbf{y}^{(k)})\}_k)$, $A_\omega^{\text{CPGD}}$ is the same as $A^{\text{GRPO}}$. (3) *Clip-filter-like weight*: when $\omega(\mathbf{x}) = \min(c_\omega, \frac{\#\{\mathbf{x} \in \mathcal{D}\}}{\#\{\mathbf{x} \in \mathcal{D}|\text{std}(\{\mathcal{R}_o(\mathbf{x}, \mathbf{y}^{(k)})\}_k) \neq 0\}})$, $c_\omega > 0$, similar weighting strategies have also been explored in concurrent work (Chu et al., 2025), with an analogous effect to online filtering (Cui et al., 2025), amplifying the gradient contribution of samples with non-zero advantage.

## 5 Experiments

**Experimental organization.** Our experiments are organized to support the paper's central claim: stable long-horizon multimodal RL training. We structure our evaluation in three tiers:

**Tier 1: Stability diagnosis.** We first demonstrate the training collapse phenomenon in existing methods (GRPO, RLOO, REINFORCE++) and show that CPGD maintains stable training throughout (Section 5.3, RL Algorithm Comparison). This directly validates our instability analysis and the effectiveness of CPGD's design.

**Tier 2: Effectiveness under controlled settings.** We evaluate CPGD's performance improvements on standard benchmarks under consistent experimental conditions (same base models, same data, same hyperparameters), demonstrating that stability translates to reliable learning (Section 5.3, Comparison with State-of-the-Art Models).

**Tier 3: Generalization evidence.** We examine whether reasoning capabilities developed on mathematics transfer to other domains (physics, chemistry, biology), providing insights into how RL training affects multimodal models (Section 5.3, MMK12 cross-domain evaluation; Discussion section).

We present our experimental setup in Section 5.1, provide an overview of baselines and benchmarks in Section 5.2, and report results in Section 5.3.

## 5.1 Experiments Setup

**RL baselines, dataset, and implementation details.** We compare CPGD with several widely used RL algorithms, including GRPO (DeepSeek-AI et al., 2025), REINFORCE++ (Ahmadian et al., 2024) and RLOO (Ahmadian et al., 2024) on the MMK12 training dataset, which contains 15,616 multimodal math problems with verified answers. We use Qwen2.5-VL-7B and Qwen2.5-VL-32B as base models, and conduct experiments with **five** random seeds[1]. Training is performed without reference constraints, and final performance is reported using the last checkpoint. Our rule-based reward consists of accuracy and format components: the former uses MathVerify to extract and compare answers, returning 1 or 0; the latter checks format compliance, returning 0.5 or 0.

We follow the prompt format from DeepSeek-R1, where reasoning steps and final answers are explicitly marked using `<think>` and `<answer>` tags, respectively. The full prompt template is provided in Table 5 (Appendix C).

For all experiments, we use the same hyperparameters: rollout and training batch sizes of 128, 8 sampled responses per prompt (temperature 1.0), a learning rate of $1e-6$, one PPO epoch, and five training episodes. No reference policy constraint is applied during training, and each run requires approximately 60 hours of computation on 8 H100 GPUs.

## 5.2 Baselines and Benchmarks

**Benchmarks, model baselines, and overall metric.** We evaluate all algorithms on six widely used benchmarks: MathVista (testmini) (Lu et al., 2024), MathVerse (testmini) (Zhang et al., 2024a), MathVision (test) (Wang et al., 2024a), OlympiadBench (EN-OE split) (He et al., 2024), WeMath (Qiao et al., 2024) and MMK12. MathVista covers visual QA, logic, algebra, and geometry; MathVerse focuses on mathematically grounded visual understanding; and MathVision extends to abstract visual reasoning. OlympiadBench targets graduate-level competition problems, while WeMath enables fine-grained diagnostic analysis via hierarchically annotated tasks. MMK12 provides 500 multiple-choice questions per subject across math, physics, chemistry, and biology for cross-domain performance evaluation.

We also include several multimodal models as baselines. We evaluate open-source models of comparable model size, trained with various strategies, including Qwen2.5-VL (Bai et al., 2023), InternVL2.5-MPO (Wang et al., 2024b), R1-OneVision (Yang et al., 2025), and OpenVLThinker (Deng et al., 2025), which collectively represent the average performance across the evaluated benchmarks. We further evaluate the leading closed-source models such as GPT-4o (Hurst et al., 2024) and OpenAI-o1 (OpenAI, 2024) to represent the most outstanding performance that the current state-of-the-art model can achieve on these benchmarks. Furthermore, to capture overall model performance across $N$ benchmarks, we define an *overall* metric by normalizing each score against a strong baseline, Qwen2.5-VL-7B: Overall $:= \frac{1}{N} \sum_{j=1}^{N} X_j / X_j^{\text{Qwen}}$, where $X_j$ and $X_j^{\text{Qwen}}$ are the model and baseline scores on benchmark $j$.

## 5.3 Main Results

**RL Algorithm Comparison** We first compare different RL algorithms under identical settings (same base model, dataset, and hyperparameters) to demonstrate the effectiveness of CPGD. As shown in Table 2, CPGD consistently outperforms GRPO, RLOO, and REINFORCE++ across all benchmarks. Compared to the base model QwenVL2.5-7B, CPGD achieves an overall improvement of 10%, with particularly strong

---

[1]Although in the field of LMs it is common to report results from a single random seed (due to high computational cost), we have run each set of experiments with five random seeds to ensure academic rigor and reproducibility.

gains on MMK12 (+24.6%), MathVista (+8.2%), and WeMath (+9.5%). Notably, CPGD demonstrates superior stability during training, avoiding the catastrophic collapse observed in other methods (see Figure 3 for training curves and Appendix E for additional experiments).

Table 2: Comparison of different RL algorithms on QwenVL2.5-7B. Mean ± std over 5 random seeds. Best in **bold**, second-best underlined.

| Model | MathVista | MathVerse | MathVision | Olypamid | WeMath | MMK12 | Overall |
|---|---|---|---|---|---|---|---|
| QwenVL2.5-7B (base) | 68.2 | 47.9 | 25.4 | 20.2 | 62.1 | 53.6 | 1.00 |
| RLOO | 70.5±1.3 | 49.0±0.9 | 20.7±1.3 | 18.9±0.4 | 67.2±1.0 | 62.1±0.7 | 1.01±0.00 |
| REINFORCE++ | 63.8±0.9 | 46.1±0.7 | 18.9±0.4 | 18.7±0.6 | 66.6±0.6 | 64.7±0.3 | 0.98±0.01 |
| GRPO | 70.7±0.8 | 50.6±0.7 | 23.0±1.6 | 19.4±0.6 | 67.2±0.6 | 65.0±0.1 | 1.04±0.01 |
| **CPGD** (ours) | **73.8**±0.5 | **51.1**±0.7 | **27.0**±0.9 | **21.2**±0.4 | **68.0**±0.6 | **66.8**±0.8 | **1.10**±0.01 |

Table 3: Performance comparison across different multimodal mathematical benchmarks. Bold indicates the top performer among all open-source models, while underline indicates the second best. All results are based on our consistent evaluation framework to ensure fair comparison.

| Model | MathVista | MathVerse | Mathvision | OlympiadBench | WeMath |
|---|---|---|---|---|---|
| **Closed-Source Models** | | | | | |
| Claude3.7-Sonnet | 66.8 | - | 41.3 | - | 72.6 |
| GPT-4o | 63.8 | 50.2 | 30.4 | 35.0 | 68.8 |
| o1 | 73.9 | 57.0 | 60.3 | - | - |
| Gemini2-flash | 70.4 | 59.3 | 41.3 | 51.0 | 71.4 |
| **Open-Source General Models** | | | | | |
| InternVL2.5-VL-8B | 64.4 | 39.5 | 19.7 | 12.3 | 53.5 |
| Qwen-2.5-VL-7B | 68.2 | 47.9 | 25.4 | 20.2 | 62.1 |
| InternVL2.5-VL-38B | 71.9 | 49.4 | 31.8 | 32.0 | 67.5 |
| Qwen-2.5-VL-32B | 71.7 | 49.9 | **40.1** | 30.0 | 69.1 |
| InternVL2.5-VL-78B | 72.3 | 51.7 | 32.2 | 31.1 | 66.3 |
| Qwen-2.5-VL-72B | **74.8** | **57.6** | 38.1 | **40.4** | 72.4 |
| **Open-Source Reasoning Models** | | | | | |
| InternVL2.5-8B-MPO | 68.9 | 35.5 | 21.5 | 7.8 | 53.5 |
| InternVL2.5-38B-MPO | 73.8 | 46.5 | 32.3 | 25.6 | 66.2 |
| QVQ-72B-Preview | 71.4 | 48.2 | 35.9 | 33.2 | 65.4 |
| Adora-7B | 73.5 | 50.1 | 23.0 | 20.1 | 64.2 |
| R1-Onevision-7B | 64.1 | 47.1 | 23.5 | 17.3 | 61.8 |
| OpenVLThinker-7B | 70.2 | 47.9 | 25.3 | 20.1 | 64.3 |
| **Ours (with CPGD)** | | | | | |
| MM-Eureka-7B | 73.8 | 51.1 | 27.0 | 21.2 | 68.0 |
| MM-Eureka-32B | **74.8** | 56.5 | 34.4 | 35.9 | **73.4** |

**Comparison with State-of-the-Art Models** Beyond the RL algorithm comparison, we evaluate MM-Eureka against state-of-the-art open-source and closed-source models. As shown in Table 3, MM-Eureka trained with CPGD demonstrates superior performance compared to similar-sized baselines. MM-Eureka-7B achieves 73.8 on MathVista, surpassing InternVL-78B and all reasoning-focused models of similar size. It also achieves 68.0 on WeMath and 51.1 on MathVerse, establishing new state-of-the-art results among 7B models. MM-Eureka-32B matches or exceeds Qwen-2.5-VL-72B across most benchmarks, achieving 74.8 on MathVista, 56.5 on MathVerse, and 73.4 on WeMath. Notably, it achieves competitive performance relative to the closed-source model Claude3.7 Sonnet across multiple benchmarks.

The results demonstrate that CPGD enables MM-Eureka to achieve competitive performance with much larger models (e.g., Qwen-72B) and even closed-source models, while maintaining stable training throughout. However, when compared to the most advanced closed-source model o1, MM-Eureka-32B still shows performance gaps on the most challenging benchmarks such as MathVision and OlympiadBench, indicating room for further improvement.

Table 4: Performance comparison across different disciplines in MMK12. Bold indicates the top performer with the open-source models, while underline indicates the second best performer within open-source models.

| Model | Mathematics | Physics | Chemistry | Biology | Avg. |
|---|---|---|---|---|---|
| **Closed-Source Models** | | | | | |
| Claude3.7-Sonnet | 57.4 | 53.4 | 55.4 | 55.0 | 55.3 |
| GPT-4o | 55.8 | 41.2 | 47.0 | 55.4 | 49.9 |
| o1 | 81.6 | 68.8 | 71.4 | 74.0 | 73.9 |
| Gemini2-flash | 76.8 | 53.6 | 64.6 | 66.0 | 65.2 |
| **Open-Source General Models** | | | | | |
| InternVL2.5-VL-8B | 46.8 | 35.0 | 50.0 | 50.8 | 45.6 |
| Qwen-2.5-VL-7B | 58.4 | 45.4 | 56.4 | 54.0 | 53.6 |
| InternVL2.5-VL-38B | 61.6 | 49.8 | 60.4 | 60.0 | 58.0 |
| Qwen-2.5-VL-32B | 71.6 | 59.4 | 69.6 | 66.6 | 66.8 |
| InternVL2.5-VL-78B | 59.8 | 53.2 | 68.0 | 65.2 | 61.6 |
| Qwen-2.5-VL-72B | **75.6** | **64.8** | 69.6 | 72.0 | 70.5 |
| **Open-Source Reasoning Models** | | | | | |
| InternVL2.5-8B-MPO | 26.6 | 25.0 | 42.4 | 44.0 | 34.5 |
| InternVL2.5-38B-MPO | 41.4 | 42.8 | 55.8 | 53.2 | 48.3 |
| QVQ-72B-Preview | 61.4 | 57.4 | 62.6 | 64.4 | 61.5 |
| Adora | 63.6 | 50.6 | 59.0 | 59.0 | 58.1 |
| R1-Onevision | 44.8 | 33.8 | 39.8 | 40.8 | 39.8 |
| OpenVLThinker-7 | 63.0 | 53.8 | 60.6 | 65.0 | 60.6 |
| **Ours** | | | | | |
| MM-Eureka-7B | 71.2 | 56.2 | 65.2 | 65.2 | 64.5 |
| MM-Eureka-32B | 74.6 | 62.0 | **75.4** | **76.8** | **72.2** |

**MMK12**   Beyond validating our model's superiority on widely-used multimodal mathematical reasoning benchmarks like MathVista, it is necessary to test its capabilities and generalization across multidisciplinary reasoning domains using questions absent from the training set (e.g., physics, chemistry, and biology). For this purpose, we employ the MMK12 dataset constructed in Section 3, which can effectively measure models' multimodal reasoning capabilities across multiple disciplines.

As shown in Table 4, MM-Eureka-32B demonstrates multidisciplinary capabilities only marginally behind o1 by 1.7%, while outperforming larger-scale models such as Qwen-2.5-VL-72B and Gemini2-Flash-Thinking. MM-Eureka-7B also surpasses several similarly-sized multimodal reasoning models, including OpenVLThinker-7B, with overall performance exceeding InternVL2.5-VL-78B and only slightly behind Qwen-2.5-VL-32B. Additionally, we observe several interesting findings: 1) Despite being trained exclusively on fill-in-the-blank questions, our models maintain strong instruction-following abilities for multiple-choice questions with improved performance. 2) Even with training solely on mathematics problems, the models exhibit enhanced capabilities in physics, chemistry, and biology. Specifically, MM-Eureka-7B shows improvements of 9.8% and 11.2% in chemistry and biology, respectively, demonstrating the remarkable generalization capacity of our straightforward RL strategy.

### 5.4 Qualitative Results

We provide qualitative comparisons in Appendix F, where representative examples across mathematics, physics, chemistry, and biology demonstrate MM-Eureka-32B's enhanced reasoning capabilities compared to its base model. These cases reveal that our model better applies known concepts and performs multi-step deduction, while the base model often shows only surface-level understanding without coherent application in problem-solving contexts.

## 6 Discussion

This section discusses insights from our experiments and acknowledges the limitations of our work.

### 6.1 Are knowledge and reasoning decoupled?

As shown in Figure 4, the distribution of correct answers reveals that MM-Eureka significantly improves accuracy on problems initially answered correctly at least once, while problems with zero correct answers remain unchanged. These results provide empirical evidence that RL training primarily improves the reliability of applying existing knowledge rather than injecting new domain knowledge. Reasoning alone cannot address missing knowledge. Our experimental results indicate that knowledge and reasoning in LLMs/VLMs can be decoupled, enabling separate learning of knowledge and reasoning. Future research should explore generalizing reasoning capabilities from structured domains like mathematics to broader applications.

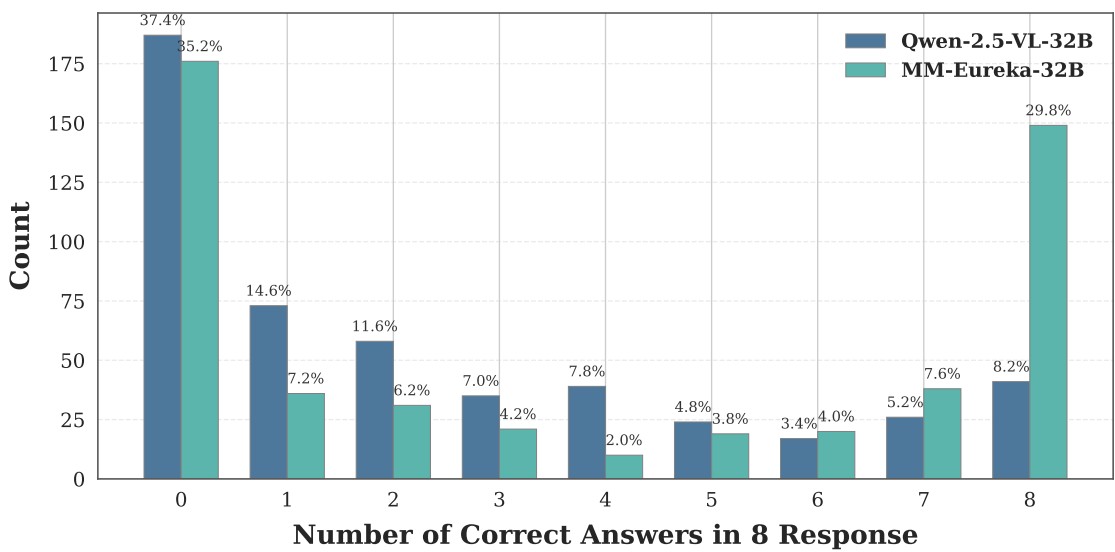

Figure 4: Distribution of correct answers across 8 responses from Qwen-2.5-VL-32B and MM-Eureka-32B on *Mathematics*.

### 6.2 RL generalizes better than SFT

We compare different post-training strategies (SFT, COT SFT, and RL) under identical settings. As shown in Appendix Table 12, RL exhibits superior generalization compared to SFT approaches, with particularly significant improvements on out-of-distribution test sets (Physics, Chemistry, Biology). While SFT methods fail to substantially improve mathematical and physical reasoning, RL training increases scores in mathematics and physics by 12.8 and 10.8 points respectively.

# 7    Conclusion

This paper addresses a fundamental training stability issue that prevents multimodal rule-based reinforcement learning from being reliably trainable. We identify the root cause as ratio-induced policy drift amplification in existing objectives, provide both theoretical and empirical evidence for this mechanism, and propose CPGD as a structural solution that removes ratios from gradient-carrying terms while maintaining proximal updates through explicit drift control. Our contributions include: (1) CPGD, a stability-oriented RL objective with theoretical guarantees; (2) an analysis of why ratio-based objectives collapse in multimodal settings; and (3) MMK12, a dataset that makes this failure mode reproducible and diagnosable. Using CPGD on MMK12, we train MM-Eureka models that demonstrate stable long-horizon training, achieving consistent improvements while maintaining stability. We believe this work establishes training stability as a first-class consideration for multimodal RL. The mechanisms we identify and the solutions we propose are not specific to a single model or benchmark, but address a broader methodological challenge in multimodal rule-based RL. By open-sourcing our pipeline, we hope to enable the community to build on these findings and further advance stable multimodal reinforcement learning.

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

# A Theoretical Analysis

## A.1 Proof for Proposition 1

**Proposition 1.** *Let $\theta_0$ be a parameter such that the importance-sampling ratio satisfies $|\frac{\pi_{\theta_0}(\mathbf{y}|\mathbf{x})}{\pi_{\theta_{old}}(\mathbf{y}|\mathbf{x})} - 1| = \epsilon$. Consider updating $\theta_0$ using either (i) the PPO-clip objective, resulting in parameter $\theta_1^{PPO}$, or (ii) the CPGD objective with $\alpha = 0$, yielding parameter $\theta_1^{CPG}$. Then, there exists a constant $\eta_{\max} > 0$ such that for any learning rate $\eta \in (0, \eta_{\max})$, the following inequality holds:*

$$\left|\frac{\pi_{\theta_1^{PPO}}(\mathbf{y}|\mathbf{x})}{\pi_{\theta_{old}}(\mathbf{y}|\mathbf{x})} - 1\right| > \left|\frac{\pi_{\theta_1^{CPG}}(\mathbf{y}|\mathbf{x})}{\pi_{\theta_{old}}(\mathbf{y}|\mathbf{x})} - 1\right| > \left|\frac{\pi_{\theta_0}(\mathbf{y}|\mathbf{x})}{\pi_{\theta_{old}}(\mathbf{y}|\mathbf{x})} - 1\right| = \epsilon.$$

*After one update step, both PPO and CPG increase the importance-sampling ratio deviation from the old policy, but PPO does so more aggressively than CPG.*

*Proof.* Consider $f(\eta) = \frac{\pi_{\theta_1^{\mathrm{CPG}}}(\mathbf{y}|\mathbf{x})}{\pi_{\theta_{old}}(\mathbf{y}|\mathbf{x})}$, where $\theta_1^{\mathrm{CPG}} = \theta_0 + \eta \nabla_\theta \hat{\mathcal{L}}_{\mathrm{CPG}}(\mathbf{x}, \mathbf{y}; \theta_0)$ is the single gradient ascent step on the empirical CPGD objective (Equation 1) without the policy drift term. The gradient of the objective takes the form:

$$\nabla_\theta \hat{\mathcal{L}}_{\mathrm{CPG}}(\mathbf{x}, \mathbf{y}; \theta) = A^{\mathrm{CPGD}}(\mathbf{x}, \mathbf{y}) \nabla_\theta \ln \pi_\theta(\mathbf{y}|\mathbf{x}).$$

Thus, for the case where $\frac{\pi_{\theta_0}(\mathbf{y}|\mathbf{x})}{\pi_{\theta_{old}}(\mathbf{y}|\mathbf{x})} = 1 + \epsilon$ and $A^{\mathrm{CPGD}}(\mathbf{x}, \mathbf{y}) > 0$, the directional derivative of $f$ at $\eta = 0$ satisfies:

$$f'(0) = \langle \frac{\nabla_\theta \pi_{\theta_0}(\mathbf{y}|\mathbf{x})}{\pi_{\theta_{old}}(\mathbf{y}|\mathbf{x})}, \nabla_\theta \hat{\mathcal{L}}_{\mathrm{CPG}}(\mathbf{x}; \theta_0) \rangle > 0.$$

Hence, there exists a constant $\eta_1 > 0$ such that for any $\eta \in (0, \eta_1)$, we have $f(\eta) > f(0)$. Similarly, when $\frac{\pi_{\theta_0}(\mathbf{y}|\mathbf{x})}{\pi_{\theta_{old}}(\mathbf{y}|\mathbf{x})} = 1 - \epsilon$ and $A^{\mathrm{CPGD}}(\mathbf{x}, \mathbf{y}) < 0$, there exists $\eta_2 > 0$ such that $f(\eta) < f(0)$ for any $\eta \in (0, \eta_2)$.

Therefore, for any $0 < \eta < \min(\eta_1, \eta_2)$, the following holds:

$$|\frac{\pi_{\theta_1^{\mathrm{CPG}}}(\mathbf{y}|\mathbf{x})}{\pi_{\theta_{old}}(\mathbf{y}|\mathbf{x})} - 1| > |\frac{\pi_{\theta_0}(\mathbf{y}|\mathbf{x})}{\pi_{\theta_{old}}(\mathbf{y}|\mathbf{x})} - 1| = \epsilon. \tag{3}$$

Next, define $g(\eta) = \frac{\pi_{\theta_1^{\mathrm{CPG}}}(\mathbf{y}|\mathbf{x})}{\pi_{\theta_{old}}(\mathbf{y}|\mathbf{x})} - \frac{\pi_{\theta_1^{\mathrm{PPO}}}(\mathbf{y}|\mathbf{x})}{\pi_{\theta_{old}}(\mathbf{y}|\mathbf{x})}$, where $\theta_1^{\mathrm{PPO}} = \theta_0 + \eta \nabla_\theta \hat{\mathcal{L}}_{\mathrm{PPO}}(\mathbf{x}, \mathbf{y}; \theta_0)$ and

$$\nabla_\theta \hat{\mathcal{L}}_{\mathrm{PPO}}(\mathbf{x}, \mathbf{y}; \theta) = A^{\mathrm{CPGD}}(\mathbf{x}, \mathbf{y}) \frac{\nabla_\theta \pi_\theta(\mathbf{y}|\mathbf{x})}{\pi_{\theta_{old}}(\mathbf{y}|\mathbf{x})}.$$

For the case where $\frac{\pi_{\theta_0}(\mathbf{y}|\mathbf{x})}{\pi_{\theta_{old}}(\mathbf{y}|\mathbf{x})} = 1 + \epsilon$ and $A^{\mathrm{CPGD}}(\mathbf{x}, \mathbf{y}) > 0$, we have:

$$g'(0) = \Big\langle \frac{\nabla_\theta \pi_{\theta_0}(\mathbf{y}|\mathbf{x})}{\pi_{\theta_{old}}(\mathbf{y}|\mathbf{x})}, A^{\mathrm{CPGD}}(\mathbf{x}, \mathbf{y}) \cdot (1 - \frac{\pi_\theta(\mathbf{y}|\mathbf{x})}{\pi_{\theta_{old}}(\mathbf{y}|\mathbf{x})}) \cdot \nabla_\theta \ln \pi_\theta(\mathbf{y}|\mathbf{x}) \Big\rangle < 0.$$

Hence, there exists a constant $\eta_3 > 0$ such that $g(\eta) < g(0)$ for any $\eta \in (0, \eta_3)$. Similarly, for the case where $\frac{\pi_{\theta_0}(\mathbf{y}|\mathbf{x})}{\pi_{\theta_{old}}(\mathbf{y}|\mathbf{x})} = 1 - \epsilon$ and $A^{\mathrm{CPGD}}(\mathbf{x}, \mathbf{y}) < 0$, there exists a constant $\eta_4 > 0$ such that $g(\eta) > g(0)$ for any $\eta \in (0, \eta_4)$.

Therefore, for any $0 < \eta < \min(\eta_3, \eta_4)$, we have

$$|\frac{\pi_{\theta_1^{\mathrm{PPO}}}(\mathbf{y}|\mathbf{x})}{\pi_{\theta_{old}}(\mathbf{y}|\mathbf{x})} - 1| > |\frac{\pi_{\theta_1^{\mathrm{CPG}}}(\mathbf{y}|\mathbf{x})}{\pi_{\theta_{old}}(\mathbf{y}|\mathbf{x})} - 1|. \tag{4}$$

Therefore, by letting $\eta_{\max} = \min(\eta_1, \eta_2, \eta_3, \eta_4)$, the proof is complete. $\square$

## A.2 Extension of Proposition 1 to CPGD ($\alpha > 0$)

Proposition 1 is stated for the case $\alpha = 0$ (i.e., CPG without policy drift regularization). A natural question is whether the result extends to the full CPGD objective with $\alpha > 0$. The answer is yes, with the following strengthened inequality:

**Proposition 2.** *Under the same conditions as Proposition 1, for sufficiently small learning rate $\eta$:*

$$\left| \frac{\pi_{\theta_1^{PPO}}(\mathbf{y}|\mathbf{x})}{\pi_{\theta_{old}}(\mathbf{y}|\mathbf{x})} - 1 \right| > \left| \frac{\pi_{\theta_1^{CPG}}(\mathbf{y}|\mathbf{x})}{\pi_{\theta_{old}}(\mathbf{y}|\mathbf{x})} - 1 \right| \geq \left| \frac{\pi_{\theta_1^{CPGD}}(\mathbf{y}|\mathbf{x})}{\pi_{\theta_{old}}(\mathbf{y}|\mathbf{x})} - 1 \right| > \epsilon.$$

*Proof.* The policy drift term $-\alpha \cdot D_{\mathrm{KL}}(\pi_{\theta_{old}} \| \pi_\theta | \mathbf{x})$ contributes an additional gradient component $-\alpha \nabla_\theta \ln \pi_\theta(\mathbf{y}|\mathbf{x})$, which always drives the policy back toward $\pi_{\theta_{old}}$. This counteracts the policy gradient update direction.

Concretely, the CPGD gradient is:

$$\nabla_\theta \hat{\mathcal{L}}_{\mathrm{CPGD}}(\mathbf{x}, \mathbf{y}; \theta) = A^{\mathrm{CPGD}}(\mathbf{x}, \mathbf{y}) \nabla_\theta \ln \pi_\theta(\mathbf{y}|\mathbf{x}) - \alpha \nabla_\theta D_{\mathrm{KL}}(\pi_{\theta_{old}} \| \pi_\theta | \mathbf{x}).$$

Compared to the CPG gradient (without the drift term), the additional $-\alpha$ term reduces the magnitude of the update in the direction that increases the ratio deviation. Since the drift term always pulls the policy toward $\pi_{\theta_{old}}$, the resulting policy deviation after one CPGD step is no larger than that of CPG, yielding the inequality $|\frac{\pi_{\theta_1^{CPG}}(\mathbf{y}|\mathbf{x})}{\pi_{\theta_{old}}(\mathbf{y}|\mathbf{x})} - 1| \geq |\frac{\pi_{\theta_1^{CPGD}}(\mathbf{y}|\mathbf{x})}{\pi_{\theta_{old}}(\mathbf{y}|\mathbf{x})} - 1|$. The remaining inequalities follow directly from Proposition 1. $\square$

## A.3 Proof for Theorem 1

**Theorem 1.** *Let $\{\pi_{\theta_k}\}_{k=0}^\infty$ denote the sequence of policies generated by the CPGD update rule: $\theta_{k+1} = \arg\max_\theta \mathcal{L}_{CPGD}(\theta; \theta_{old})$ where the advantage function is competed as $A^{CPGD}(\mathbf{x}, \mathbf{y}) = \mathcal{R}_o(\mathbf{x}, \mathbf{y})$. Then, $\pi_{\theta_{k+1}}$ is better than $\pi_{\theta_k}$, i.e., $\eta(\theta_{k+1}) \geq \eta(\theta_k)$, where $\eta(\theta) := \mathbb{E}_{\mathbf{x} \sim \mathcal{D}, \mathbf{y} \sim \pi_{\theta_{old}}(\cdot|\mathbf{x})}[\mathcal{R}_o(\mathbf{x}, \mathbf{y})]$.*

*Proof.* First, denote $\mathcal{L}_{\mathrm{CPGD}}(\theta; \theta_k) = \mathbb{E}_{\mathbf{x} \sim \mathcal{D}}[g(\theta; \theta_k, \mathbf{x})]$, and rewrite $g$ as

$$g(\theta; \theta_k, \mathbf{x}) = \mathbb{E}_{\mathbf{y} \sim \pi_{\theta_k}(\cdot|\mathbf{x})}\left[\mathcal{R}_o(\mathbf{x}, \mathbf{y}) \ln \frac{\pi_\theta(\mathbf{y}|\mathbf{x})}{\pi_{\theta_k}(\mathbf{y}|\mathbf{x})}\right] - \alpha D_{\mathrm{KL}}(\pi_{\theta_k}, \pi_\theta | \mathbf{x})$$

$$- \mathbb{E}_{\mathbf{y} \sim \pi_{\theta_k}(\cdot|\mathbf{x})}\left[\mathrm{ReLU}\left(\left[\ln \frac{\pi_\theta(\mathbf{y}|\mathbf{x})}{\pi_{\theta_k}(\mathbf{y}|\mathbf{x})} - \mathrm{clip}\left(\ln \frac{\pi_\theta(\mathbf{y}|\mathbf{x})}{\pi_{\theta_k}(\mathbf{y}|\mathbf{x})}, \ln(1-\epsilon), \ln(1+\epsilon)\right)\right]\mathcal{R}_o(\mathbf{x}, \mathbf{y})\right)\right].$$

Here, we omit the baseline $\mathbb{E}_{\mathbf{y} \sim \pi_{\theta_k}(\cdot|\mathbf{x})}[\mathcal{R}_o(\mathbf{x}, \mathbf{y})]$. Then, denoting $\theta_{k+1}$ the point such that $\mathcal{L}_{\mathrm{CPGD}}(\theta_{k+1}; \theta_k) \geq \mathcal{L}_{\mathrm{CPGD}}(\theta_k; \theta_k)$, we obtain

$$\mathbb{E}_{\mathbf{y} \sim \pi_{\theta_{k+1}}(\cdot|\mathbf{x})}\left[\mathcal{R}_o(\mathbf{x}, \mathbf{y})\right] - \mathbb{E}_{\mathbf{y} \sim \pi_{\theta_k}(\cdot|\mathbf{x})}\left[\mathcal{R}_o(\mathbf{x}, \mathbf{y})\right]$$

$$= \mathbb{E}_{\mathbf{y} \sim \pi_{\theta_k}(\cdot|\mathbf{x})}\left[\left(\frac{\pi_{\theta_{k+1}}(\mathbf{y}|\mathbf{x})}{\pi_{\theta_k}(\mathbf{y}|\mathbf{x})} - 1\right)\mathcal{R}_o(\mathbf{x}, \mathbf{y})\right]$$

$$\geq \mathbb{E}_{\mathbf{y} \sim \pi_{\theta_k}(\cdot|\mathbf{x})}\left[\ln \frac{\pi_{\theta_{k+1}}(\mathbf{y}|\mathbf{x})}{\pi_{\theta_k}(\mathbf{y}|\mathbf{x})} \cdot \mathcal{R}_o(\mathbf{x}, \mathbf{y})\right]$$

$$= g(\theta_{k+1}; \theta_k, \mathbf{x}) - g(\theta_k; \theta_k, \mathbf{x}) + \alpha D_{\mathrm{KL}}(\pi_{\theta_k}, \pi_{\theta_{k+1}} | \mathbf{x})$$

$$+ \mathbb{E}_{\mathbf{y} \sim \pi_{\theta_k}(\cdot|\mathbf{x})}\left[\mathrm{ReLU}\left(\left[\ln \frac{\pi_{\theta_{k+1}}(\mathbf{y}|\mathbf{x})}{\pi_{\theta_k}(\mathbf{y}|\mathbf{x})} - \mathrm{clip}\left(\ln \frac{\pi_{\theta_{k+1}}(\mathbf{y}|\mathbf{x})}{\pi_{\theta_k}(\mathbf{y}|\mathbf{x})}, \ln(1-\epsilon), \ln(1+\epsilon)\right)\right]\mathcal{R}_o(\mathbf{x}, \mathbf{y})\right)\right].$$

Denoting the overall expected return by $\eta(\pi_\theta) = \mathbb{E}_{\mathbf{x} \sim \mathcal{D}, \mathbf{y} \sim \pi_\theta(\cdot|\mathbf{x})}[\mathcal{R}_o(\mathbf{x}, \mathbf{y})]$, we integrate over $\mathbf{x}$ to conclude

$$\eta(\theta_{k+1}) - \eta(\theta_k) \geq \alpha \mathbb{E}_{\mathbf{x} \sim \mathcal{D}}\left[D_{\mathrm{KL}}(\pi_{\theta_k}, \pi_{\theta_{k+1}} | \mathbf{x})\right] \overset{\text{Pinsker inequality}}{\geq} \frac{\alpha}{2}\|\pi_{\theta_{k+1}} - \pi_{\theta_k}\|_1^2.$$

Therefore, we have $\eta(\theta_{k+1}) \geq \eta(\theta_k)$. $\square$

# B    MMK12 Dataset Details

## B.1    Construction Process

We collect diverse K12-level multimodal math problems from multiple sources, including Chinese mathematics textbooks and examination papers across elementary, middle, and high school levels. The construction pipeline involves:

**Data Collection:** We gather multimodal mathematics problems with corresponding answers from Chinese educational materials. For the training set, we focus on fill-in-the-blank problems to minimize false positives during training. For the evaluation set, we select multiple-choice questions to facilitate reliable and efficient evaluation.

**Translation and Refinement:** Questions, answers, and CoT (Chain-of-Thought) processes are translated from Chinese to English with the assistance of LLMs. We carefully refine the translations to maintain mathematical accuracy and clarity.

**Answer Verification:** We use Math-Verify[2] to parse and verify answers, ensuring data reliability for RL training. This step is crucial for maintaining high-quality training signals.

## B.2    Problem Categories

MMK12 encompasses mathematical problems across various knowledge domains:

- **Function Reasoning**: This task requires models to understand function concepts, analyze function graphs and expressions, and apply function properties to solve problems. This type of reasoning develops the model's ability to comprehend abstract mathematical concepts, fostering its capability to identify function characteristics, determine critical points, and analyze function behavior.

- **Geometric Reasoning**: This task involves applying spatial relationships, geometric theorems, and properties of shapes. Through geometric reasoning training, models enhance their spatial visualization, logical deduction, and formalization abilities for geometry problems, enabling them to solve complex problems in both plane and solid geometry.

- **Pattern Reasoning**: This type of task focuses on understanding flow diagrams and recognizing patterns in visual sequences. Models need to discover patterns, predict rule-based changes, or understand logical relationships in visual content. This task examines the model's pattern recognition abilities, inductive reasoning skills, and visual logical thinking.

# C    Details of Experiments

## C.1    Basic Settings

The training is conducted using GRPO as the base RL algorithm. To support multimodal inputs, we develop a custom RL framework built on OpenRLHF (Hu et al., 2024). Following DeepSeek-R1, we also adopt the simple rule-based reward function rather than using outcome or process reward models, thereby alleviating reward hacking (Gao et al., 2022). Specifically, we use two types of rewards: accuracy reward and format reward. The former uses Math-Verify to extract the answer from model responses and compare it with the reference one, returning 1 or 0 based on correctness; the latter checks whether the response follows the specified format (`<think>`...`</think><answer>`...`</answer>`), returning 0.5 or 0 based on compliance. We find that this simple and sparse reward is sufficient to significantly improve the model's multimodal reasoning ability.

Table 5: Prompt setting.

---

**SYSTEM:** Solve the question. The user asks a question, and you solves it. You first thinks about the reasoning process in the mind and then provides the user with the answer. The answer is in latex format and wrapped in $...$. The final answer must be wrapped using the \boxed{} command. The reasoning process and answer are enclosed within <think></think> and <answer></answer> tags, respectively, i.e., <think>Since $1+1 = 2$, so the answer is 2. </think><answer>The answer is $\boxed{2}$ </answer>, which means the final answer assistant's output should start with <answer> and end with </answer>.

**USER:** <image>{{question}}

---

## C.2 Prompt Setting

We follow the prompt format from DeepSeek-R1, where reasoning steps and final answers are explicitly marked using `<think>` and `<answer>` tags, respectively. The full prompt template is provided in Table 5.

## C.3 Answer Verification Details

We adopt a two-component reward function that combines an **accuracy reward** and a **format reward**:

**Accuracy Reward.** We use Math-Verify to parse and compare model-generated answers against reference solutions. Math-Verify supports a wide range of mathematical expressions including numerical values, fractions, symbolic expressions, and LaTeX-formatted answers (e.g., `\boxed{...}`). The parser first extracts the answer from the model's response (specifically from within the `<answer>` tags), then normalizes both the predicted and reference answers into a canonical form before performing equivalence comparison. The accuracy reward returns 1 if the answers match and 0 otherwise.

**Format Reward.** We additionally check whether the model's response follows the prescribed structured format: `<think>...</think><answer>...</answer>`. Specifically, we verify that: (1) the response contains exactly one `<think>...</think>` block followed by exactly one `<answer>...</answer>` block, and (2) the `<answer>` block contains a `\boxed{...}` expression. The format reward returns 0.5 if the format is correct and 0 otherwise.

## C.4 Hyperparameters

For all experiments, we use the same hyperparameters: rollout and training batch sizes of 128, 8 sampled responses per prompt (temperature 1.0), a learning rate of $1e-6$, one PPO epoch, and five training episodes. These choices follow established practice in recent concurrent works: the learning rate of $1e-6$ is consistent with Qwen-based RL training in (Hu et al., 2025; Peng et al., 2025), and the batch size of 128 follows common OpenRLHF-style implementations (Hu et al., 2024; Peng et al., 2025). No reference policy constraint is applied during training, final performance is reported using the last checkpoint, and each run requires approximately 60 hours of computation on 8 H100 GPUs.

## C.5 Details of Benchmarks

We evaluate all algorithms on six widely used benchmarks: MathVista (testmini) (Lu et al., 2024), MathVerse (testmini) (Zhang et al., 2024a), MathVision (test) (Wang et al., 2024a), OlympiadBench (EN-OE split) (He et al., 2024), WeMath (Qiao et al., 2024) and MMK12. MathVista covers visual QA, logic, algebra, and geometry; MathVerse focuses on mathematically grounded visual understanding; and MathVision extends to abstract visual reasoning. OlympiadBench targets graduate-level competition problems, while WeMath enables fine-grained diagnostic analysis via hierarchically annotated tasks. MMK12 provides 500 multiple-

---

[2]https://github.com/huggingface/Math-Verify

choice questions per subject across math, physics, chemistry, and biology for cross-domain performance evaluation.

## C.6 Addition Experiment on Other Model Backbones

Table 6: Comparisons of CPGD and GRPO on Internvl2.5 and QwenVL2.5-32B across all benchmarks.

| Model | MathVista | MathVerse | MathVision | Olypamid | WeMath | MMK12 | Overall |
|---|---|---|---|---|---|---|---|
| InternVL2.5 | 64.4 | 39.5 | 15.8 | 12.3 | 49.4 | 46.5 | 1.00 |
| InternVL2.5-GRPO | 66.8±0.6 | **41.1**±0.7 | 20.1±0.5 | 9.9±0.5 | 53.8±0.5* | 48.2±0.4 | 1.05±0.01 |
| InternVL2.5-CPGD | **68.8**±0.6 | 41.0±0.5* | **22.2**±0.7 | **13.3**±0.2 | **54.0**±0.3 | **49.2**±0.3 | **1.12**±0.01 |
| QwenVL2.5-32B | 71.7 | 49.9 | 40.1 | 30.0 | 69.1 | 66.8 | 1.00 |
| QwenVL2.5-32B-GRPO | 74.0±0.3 | 55.9±0.6 | 30.6±1.1* | 35.7±0.6 | 71.4±1.2 | 73.1±1.1 | 1.04±0.00 |
| QwenVL2.5-32B-CPGD | **75.5**±0.3 | **58.0**±0.5 | **31.8**±0.4 | **40.9**±0.3 | **74.2**±0.5 | **76.1**±0.3 | **1.10**±0.00 |

Table 7: Performance of CPGD on Qwen3-VL-8B across multimodal mathematical benchmarks.

| Model | MathVista | MathVerse | MathVision | OlympiadBench | WeMath | MMK12 | Overall |
|---|---|---|---|---|---|---|---|
| Qwen3-VL-8B-Instruct | 77.2 | 62.1 | 53.9 | 34.0 | 70.5 | 67.0 | 1.00 |
| Qwen3-VL-8B-CPGD | **81.5** | **69.0** | **59.6** | **35.6** | **83.2** | **78.2** | **1.15** |

Table 8: Comparisons of CPGD and GRPO on Qwen3-8B across all benchmarks (avg@8).

| Model | AIME2024 | AIME2025 | MATH-500 | Overall |
|---|---|---|---|---|
| Qwen3-8B | 10.5 | 10.4 | 60.1 | 1.00 |
| Qwen3-8B-GRPO | 23.6±0.6 | 22.5±1.2 | 72.4±0.9 | 1.87±0.05 |
| Qwen3-8B-CPGD | **28.4**±0.4 | **26.2**±0.9 | **75.6**±0.2 | **2.16**±0.02 |

Tables 6, 7, and 8 present detailed comparisons for CPGD on InternVL2.5-8B, QwenVL2.5-32B, Qwen3-VL-8B, and Qwen3-8B (text-only). For the multimodal experiments, the training pipeline and hyperparameters are kept exactly the same as those used on QwenVL2.5-8B. As shown in Table 7, training Qwen3-VL-8B-Instruct with MMK12 and CPGD yields consistent improvements across all benchmarks, demonstrating that our data-algorithm combination generalizes effectively to newer model backbones. For Qwen3-8B, we instead adopt the following hyperparameters: train batch size of 2048, rollout batch size of 512, and 16 responses per prompt (temperature 1.0), a learning rate of $1e-6$, one PPO epoch, and five training episodes. The train dataset we use is DAPO-17k-math (Yu et al., 2025). Furthermore, since Qwen3-8B demonstrates strong instruction-following ability, we only apply the MathVerify-based accuracy reward without using the format reward.

## C.7 Ablation Study on Importance-Sampling Ratio

Table 9 presents ablation studies on reintroducing importance-sampling ratios into CPGD. Methods employing a global clip function achieve performance comparable to the baseline, while dual clip variants show degradation. See Appendix D for detailed discussion.

## C.8 Component Ablation

To isolate the contribution of each component in CPGD, we conduct two ablation studies: one removing individual components (Table 10) and one comparing different weighting strategies (Table 11).

Table 9: Results of ablation studies on importance-sampling ratio. Top performer is in **bold**.

| Algorithm | MathVista | MathVerse | MathVision | Olypamid | WeMath | MMK12 |
|---|---|---|---|---|---|---|
| CPGD | **73.8**±0.5 | 51.1±0.7* | **27.0**±0.9 | **21.2**±0.4 | **68.0**±0.6 | **66.8**±0.8 |
| **Ablation on the importance-sampling (IS) ratio (using STD weight)** | | | | | | |
| CPGD w/ global clip IS | 73.2±0.3 | **51.5**±0.6 | 26.1±0.6 | 21.0±0.4* | 67.5±0.2 | 65.5±0.4 |
| CPGD w/ dual clip IS | 71.4±0.1 | 49.6±0.1 | 25.9±0.7 | 20.4±0.2 | 65.7±0.4 | 64.5±0.4 |

Table 10: Component ablation. PG = basic policy gradient (no clip, no drift); PGD = PG + policy drift; CPG = PG + clip mechanism; CPGD = CPG + policy drift (full method).

| Algorithm | MathVista | MathVerse | MathVision | Olypamid | WeMath | MMK12 | Overall |
|---|---|---|---|---|---|---|---|
| **CPGD (STD weight)** | **74.0** | 50.6 | **28.3** | **21.4** | 68.3 | 65.3 | **1.11** |
| PG | 67.8 | 42.0 | 22.5 | 8.0 | 58.6 | 65.9 | 0.89 |
| PGD | 64.2 | 41.1 | 20.8 | 7.5 | 58.3 | 67.3 | 0.86 |
| CPG | 72.7 | **52.3** | 27.6 | 20.8 | **70.7** | **66.2** | 1.11 |

The results reveal a clear hierarchy of component importance: (1) **Clip mechanism** is the most critical — removing it (CPG → PG) causes a dramatic drop from 1.11 to 0.89 overall, confirming that clipping prevents response length collapse and ensures proximal updates. (2) **Weighted advantage** significantly improves learning — unprocessed rewards yield only 0.85, while proper weighting achieves 1.09–1.11. (3) **Policy drift** provides secondary stabilization — its primary role is as a safeguard against extreme ratio deviations rather than a primary performance driver.

## C.9 KL Divergence Estimators

Policy drift leverages the decomposability of the logarithm function and applies the following transformations:

$$D_{\mathrm{KL}}(\pi_{\theta_{old}}, \pi_\theta | \mathbf{x}) = \mathbb{E}_{\mathbf{y} \sim \pi_{\theta_{old}}(\cdot|\mathbf{x})} \left[ \ln \frac{\pi_{\theta_{old}}(\mathbf{y}|\mathbf{x})}{\pi_\theta(\mathbf{y}|\mathbf{x})} \right] = \mathbb{E}_{\mathbf{y} \sim \pi_{\theta_{old}}(\cdot|\mathbf{x})} \left[ \sum_{i=1}^{|\mathbf{y}|} \ln \frac{\pi_{\theta_{old}}(y_i|\mathbf{x}, \mathbf{y}_{<i})}{\pi_\theta(y_i|\mathbf{x}, \mathbf{y}_{<i})} \right] \quad (5)$$

$$= \mathbb{E}_{\mathbf{y} \sim \pi_{\theta_{old}}(\cdot|\mathbf{x})} \left[ \sum_{i=1}^{|\mathbf{y}|} \left( \frac{\pi_\theta(y_i|\mathbf{x}, \mathbf{y}_{<i})}{\pi_{\theta_{old}}(y_i|\mathbf{x}, \mathbf{y}_{<i})} - 1 - \ln \frac{\pi_\theta(y_i|\mathbf{x}, \mathbf{y}_{<i})}{\pi_{\theta_{old}}(y_i|\mathbf{x}, \mathbf{y}_{<i})} \right) \right]. \quad (6)$$

Equations 5 and 6 correspond to the $k_1$ and $k_3$ estimators of the KL divergence. However, both have drawbacks. The $k_1$ estimator yields a one-side gradient direction, regardless of how far the policy has drifted, leading to wrong correction. The $k_3$ estimator provides a directionally adaptive gradient, but can become numerically unstable when the policy ratio is large:

$$\nabla_\theta \ln \frac{\pi_{\theta_{old}}(y_i|\mathbf{x}, \mathbf{y}_{<i})}{\pi_\theta(y_i|\mathbf{x}, \mathbf{y}_{<i})} = -\nabla_\theta \ln \pi_\theta(y_i|\mathbf{x}, \mathbf{y}_{<i}),$$

$$\nabla_\theta \left( \frac{\pi_\theta(y_i|\mathbf{x}, \mathbf{y}_{<i})}{\pi_{\theta_{old}}(y_i|\mathbf{x}, \mathbf{y}_{<i})} - 1 - \ln \frac{\pi_\theta(y_i|\mathbf{x}, \mathbf{y}_{<i})}{\pi_{\theta_{old}}(y_i|\mathbf{x}, \mathbf{y}_{<i})} \right) = \left( \frac{\pi_\theta(y_i|\mathbf{x}, \mathbf{y}_{<i})}{\pi_{\theta_{old}}(y_i|\mathbf{x}, \mathbf{y}_{<i})} - 1 \right) \nabla_\theta \ln \pi_\theta(y_i|\mathbf{x}, \mathbf{y}_{<i}).$$

To address this, we propose a clipped gradient variant of $k_3$ that retains its correctness of correction direction while improving stability. Specifically, our estimator $\mathcal{E}^i_{\theta_{old}, \theta}$ has the following gradient:

$$\nabla_\theta \mathcal{E}^i_{\theta_{old}, \theta}(\mathbf{x}, \mathbf{y}) = \min \left( \frac{\mathrm{sg}(\pi_\theta(y_i|\mathbf{x}, \mathbf{y}_{<i}))}{\pi_{\theta_{old}}(y_i|\mathbf{x}, \mathbf{y}_{<i})} - 1, c \right) \cdot \nabla_\theta \ln \pi_\theta(y_i|\mathbf{x}, \mathbf{y}_{<i}).$$

This ensures that: (1) When the policy ratio is moderate, the behavior matches the $k_3$ estimator; (2) When the ratio exceeds the threshold $c + 1$, the gradient is capped but still points in the correct corrective direction. In summary, our estimator uniquely combines correct corrective direction and numerical stability, outperforming both $k_1$ and $k_3$ estimators in controlling policy drift effectively.

Table 11: Weighting factor ablation for CPGD.

| Weighting | MathVista | MathVerse | MathVision | Olypamid | WeMath | MMK12 | Overall |
|---|---|---|---|---|---|---|---|
| Unprocessed rewards | 69.1 | 40.2 | 21.8 | 3.5 | 59.7 | 67.2 | 0.85 |
| Equal weight | 73.1 | 51.1 | 27.2 | 20.8 | 67.9 | 65.8 | 1.09 |
| Clip-filter-like weight | 73.4 | 51.4 | 25.9 | **21.5** | **70.2** | **67.3** | 1.10 |
| STD weight | **74.0** | **50.6** | **28.3** | 21.4 | 68.3 | 65.3 | **1.11** |

## D  Discussion Details

### D.1  Discussion on Importance Sampling

Importance sampling corrects distribution mismatch between the behavior and learned policies, improving sample efficiency. We omit the ratio to reduce variance, but do not recommend discarding it entirely. Our decision is based on two key observations: (1) the clipping fraction is only ~1% (Figure 3), and (2) we use a single PPO epoch. Thus, we argue that the importance-sampling ratio should be reintroduced when the clipping fraction is larger or multiple PPO epochs are applied:

$$A_\omega^{\mathrm{CPGD}}(\mathbf{x}, \mathbf{y}) \leftarrow \mathcal{C}\Big(\frac{\mathrm{sg}(\pi_\theta(y_i|\mathbf{x}, \mathbf{y}_{<i}))}{\pi_{\theta_{old}}(y_i|\mathbf{x}, \mathbf{y}_{<i})}\Big) A_\omega^{\mathrm{CPGD}}(\mathbf{x}, \mathbf{y}),$$

where $\mathcal{C}(\cdot)$ denotes an arbitrary truncation function, used to control variance by bounding the importance weights. We evaluate two specific forms of $\mathcal{C}(\cdot)$:

$$\text{dual clip:} \quad \mathcal{C}\Big(\frac{\pi_\theta(y_i|\mathbf{x}, \mathbf{y}_{<i})}{\pi_{\theta_{old}}(y_i|\mathbf{x}, \mathbf{y}_{<i})}\Big) = \mathrm{clip}_0^{1+\epsilon}\Big(\frac{\pi_\theta(y_i|\mathbf{x}, \mathbf{y}_{<i})}{\pi_{\theta_{old}}(y_i|\mathbf{x}, \mathbf{y}_{<i})}\Big) \cdot \mathbf{1}_{A_\omega^{\mathrm{CPGD}}(\mathbf{x}, \mathbf{y}) \geq 0}$$

$$+ \mathrm{clip}_{1-\epsilon}^c\Big(\frac{\pi_\theta(y_i|\mathbf{x}, \mathbf{y}_{<i})}{\pi_{\theta_{old}}(y_i|\mathbf{x}, \mathbf{y}_{<i})}\Big) \cdot \mathbf{1}_{A_\omega^{\mathrm{CPGD}}(\mathbf{x}, \mathbf{y}) < 0},$$

$$\text{global clip:} \quad \mathcal{C}\Big(\frac{\pi_\theta(y_i|\mathbf{x}, \mathbf{y}_{<i})}{\pi_{\theta_{old}}(y_i|\mathbf{x}, \mathbf{y}_{<i})}\Big) = \mathrm{clip}_{1-\epsilon}^{1+\epsilon}\Big(\frac{\pi_\theta(y_i|\mathbf{x}, \mathbf{y}_{<i})}{\pi_{\theta_{old}}(y_i|\mathbf{x}, \mathbf{y}_{<i})}\Big).$$

The introduction of the dual clip function enables CPGD to share nearly identical gradients with PPO with dual clip mechanism—except in cases where the advantage is negative and the importance-sampling ratio exceeds $c$. In contrast, the global clip function constrains all policy ratios strictly within the range $[1-\epsilon, 1+\epsilon]$. We empirically compare these variants and report their performance in Table 9. Methods employing a global clip function achieve performance comparable to those omitting the importance-sampling ratio, likely due to the stricter truncation applied. In contrast, approaches using a dual clip function exhibit notable performance degradation. These results indicate that more stable integration of the importance-sampling ratio remains an open research problem.

### D.2  Forward KL Divergence vs. Reverse KL Divergence

Our policy drift is based on the *forward KL divergence* $D_{\mathrm{KL}}(\pi_{old}, \pi)$, which is also used in PPO-KL (Schulman et al., 2017). However, our approach differs fundamentally in how this KL is estimated and applied. PPO-KL typically uses the $k_1$ estimator or a better $k_3$ estimator, while we introduce a novel gradient-based estimator (Section 4.4) that offers both correct corrective gradients and numerical stability, overcoming the limitations of existing estimators like $k_1$ (incorrect gradient direction) and $k_3$ (instability).

*Reverse KL divergence* $D_{\mathrm{KL}}(\pi, \pi_{old})$ is more commonly used in related work due to its connection to mirror descent and stronger convergence guarantees (Geist et al., 2019; Shani et al., 2020). Although these two KL forms are different in how they are calculated, they often lead to similar results in practice (Hsu et al., 2020). Their gradient difference is typically small during training, especially when the policy ratio is close to 1, which is common in stable learning regimes:

$$\nabla_\theta D_{\mathrm{KL}}(\pi_\theta, \pi_{\theta_{old}}|\mathbf{x}) - \nabla_\theta D_{\mathrm{KL}}(\pi_{\theta_{old}}, \pi_\theta|\mathbf{x}) \approx \mathbb{E}_{\mathbf{y} \sim \pi_{\theta_{old}}(\cdot|\mathbf{x})}\Big[\frac{1}{2}\Big(\frac{\pi_\theta(\mathbf{y}|\mathbf{x})}{\pi_{\theta_{old}}(\mathbf{y}|\mathbf{x})} - 1\Big)^2 \nabla_\theta \ln \pi_\theta(\mathbf{y}|\mathbf{x})\Big].$$

This approximation holds because $x \ln x \approx x - 1 + \frac{1}{2}(x-1)^2$ when $x$ is close to 1. Despite their similarity, we prefer forward KL for two main reasons: (1) It avoids importance sampling, which reverse KL requires; and (2) It can be cleanly split into per-token terms (see Equation 6), which is not possible with reverse KL due to the importance weights.

### D.3    Training Methods Comparison

Table 12: Performance comparison of different training methods on MMK12. In terms of both enhancing mathematical capabilities and generalizing to other disciplines, RL significantly outperforms SFT or COT SFT.

| Model | Mathematics | Physics | Chemistry | Biology | Avg. |
|---|---|---|---|---|---|
| Qwen-2.5-VL-7B | 58.4 | 45.4 | 56.4 | 54.0 | 53.5 |
| + SFT | 56.6 | 50.0 | 63.2 | 61.2 | 57.7 |
| + COT SFT | 59.2 | 46.0 | 62.2 | 61.2 | 57.1 |
| + RL | **71.2** | **56.2** | **65.2** | **65.0** | **64.5** |

We maintain consistent settings with our RL training to compare different post-training strategies including SFT and COT SFT. Using the ms-swift framework (Zhao et al., 2024), we conduct both SFT and COT SFT training for 10 epochs with identical data.

## E    Additional Collapse Studies

We further verify that the observed training collapse generalizes beyond MMK12 and is robust to hyperparameter choices. Figure 5 summarizes the results.

**Hyperparameter Sensitivity.** We ran additional sensitivity checks for GRPO to verify that the observed training collapse is not an artifact of a particular hyperparameter configuration. As shown in Figure 5a, collapse consistently occurs across a range of batch sizes and learning rates: at around steps 800 / 500 / 400 / 300 for batch sizes 64 / 128 / 256 / 512 (with learning rate $1 \times 10^{-6}$), and at around steps 800 / 500 for learning rates $5 \times 10^{-7}$ / $1 \times 10^{-6}$ (with batch size 128). These results suggest that the instability is not due to a single hyperparameter choice, but is a robust property of the ratio-based optimization dynamics we analyze.

**Collapse on other dataset.** To verify that the collapse phenomenon generalizes beyond MMK12, we conduct a new collapse study on the DeepVision dataset. We randomly sample a fill-in-the-blank subset with the same scale as MMK12 and compare GRPO and CPGD under matched training settings. As shown in Figure 5b, the same qualitative conclusion holds: GRPO still exhibits mid-training collapse (with training reward peaking early and then declining steadily), while CPGD remains stable and continues to improve throughout training. This strengthens our claim that the failure mode is not specific to MMK12 but is an inherent property of ratio-based policy gradient methods.

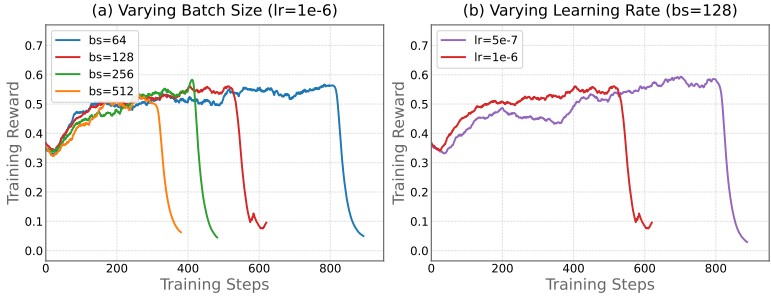 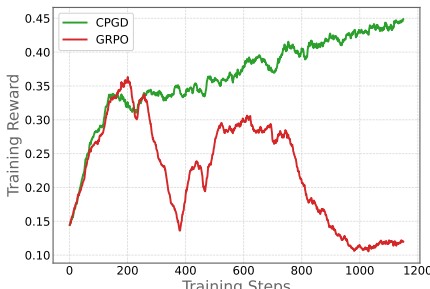

(a) GRPO training reward under varying hyperparameters on MMK12. **Left:** varying batch size (lr=$1\times10^{-6}$). **Right:** varying learning rate (bs=128).

(b) GRPO vs. CPGD on DeepVision. GRPO collapses; CPGD remains stable.

Figure 5: Additional collapse studies. Collapse occurs across all tested GRPO configurations and generalizes to the DeepVision dataset, while CPGD remains stable throughout training.

## F   Qualitative Analysis

We present representative examples comparing MM-Eureka-32B with its base model Qwen2.5-VL-32B-Instruct across four subjects (Figures 6–9). After reinforcement learning, MM-Eureka-32B demonstrates significantly enhanced reasoning: it applies known concepts through multi-step deduction to arrive at correct answers, whereas the base model often recalls relevant facts but fails to apply them coherently.

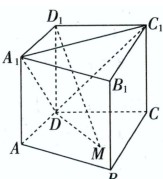

**Question:**
As shown in the figure, in the straight square column $ABCD - A_1B_1C_1D_1$, $\angle ADC = 90°$, and $AA_1 = AD = DC = 1$, $M \in$ plane $ABCD$, when $D_1M \perp$ plane $A_1C_1D$, $D_1M = $____.

**Answer:** $\sqrt{3}$

---

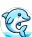 **Qwen2.5-VL-32B-Instruct**

To solve the problem, we need to determine the length of $D_1M$, where $M$ is a point in the plane $ABCD$ such that $D_1M \perp$ plane $A_1C_1D$.
...
**2. Coordinates of Points:** Place the prism in a 3D Cartesian coordinate system:

$$A = (0,0,0), \quad B = (1,0,0), \quad C = (1,1,0), \quad D = (0,1,0)$$

$$A_1 = (0,0,1), \quad B_1 = (1,0,1), \quad C_1 = (1,1,1), \quad D_1 = (0,1,1)$$

Let $M = (x, y, 0)$, where $M$ is in the plane $ABCD$.

**3. Vector Representation:** $\overrightarrow{D_1M} = (x - 0, y - 1, 0 - 1) = (x, y - 1, -1)$

**4. Condition for Perpendicularity:** For $D_1M \perp$ plane $A_1C_1D$, the vector $\overrightarrow{D_1M}$ must be perpendicular to the normal vector of the plane $A_1C_1D$.

First, find two non-parallel vectors in the plane $A_1C_1D$:
...

Therefore, the length of $D_1M$ is $\boxed{\dfrac{\sqrt{6}}{2}}$.

---

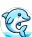 **MM-EUREKA-32B**

To solve the problem, we need to determine the length of $D_1M$ when $D_1M \perp$ plane $A_1C_1D$. Let's break the problem into steps:
...
**2. Determination of the plane $A_1C_1D$:**
- The points $A_1 = (1,0,1)$, $C_1 = (0,1,1)$, and $D = (0,0,0)$ lie in the plane $A_1C_1D$.

- To find the normal vector to the plane $A_1C_1D$, we use the cross product of two vectors in the plane:

$$\overrightarrow{A_1D} = (1-0, 0-0, 1-0) = (1,0,1), \quad \overrightarrow{C_1D} = (0-0, 1-0, 1-0) = (0,1,1).$$

- The normal vector $\mathbf{n}$ to the plane is given by the cross product:

$$\mathbf{n} = \overrightarrow{A_1D} \times \overrightarrow{C_1D} = \begin{vmatrix} \mathbf{i} & \mathbf{j} & \mathbf{k} \\ 1 & 0 & 1 \\ 0 & 1 & 1 \end{vmatrix} = \mathbf{i}(0 \cdot 1 - 1 \cdot 1) - \mathbf{j}(1 \cdot 1 - 1 \cdot 0) + \mathbf{k}(1 \cdot 1 - 0 \cdot 0) = (-1, -1, 1).$$

- Thus, the normal vector to the plane $A_1C_1D$ is $\mathbf{n} = (-1, -1, 1)$.

**3. Condition for $D_1M \perp$ plane $A_1C_1D$:**
- Since $D_1M \perp$ plane $A_1C_1D$, the vector $\overrightarrow{D_1M}$ must be parallel to the normal vector $\mathbf{n} = (-1, -1, 1)$.

- Let $M = (x, y, 0)$ be a point in the plane $ABCD$ (since $M \in$ plane $ABCD$, the $z$-coordinate of $M$ is 0).

- The vector $\overrightarrow{D_1M} = (x - 0, y - 0, 0 - 1) = (x, y, -1)$.
...
**4. Calculation of $D_1M$:**
- The vector $\overrightarrow{D_1M} = (1 - 0, 1 - 0, 0 - 1) = (1, 1, -1)$.

- The length of $D_1M$ is given by the magnitude of $\overrightarrow{D_1M}$: $|\overrightarrow{D_1M}| = \sqrt{1^2 + 1^2 + (-1)^2} = \sqrt{1 + 1 + 1} = \sqrt{3}$.

<answer>The answer is $\boxed{\sqrt{3}}$. </answer>

---

Figure 6: Example from mathematics. Red highlights Qwen2.5-VL-32B-Instruct's reasoning errors. Green shows MM-Eureka-32B's correct reasoning.

**Question:**
As shown in the figure, a lightweight spring is placed vertically on a horizontal surface with a mass of $6kg$ object $A$ resting on its upper end, in a stationary state. Now, a mass of $4kg$ object $B$ is gently placed on top of object $A$, and the spring remains within its elastic limit. Taking $g$ as $10m/s^2$, at the moment $B$ is placed, the magnitude of the force between $A$ and $B$ is (   ) A. $8N$

B. $18N$     C. $24N$     D. $40N$

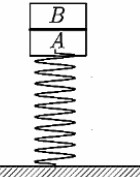

**Answer:** $C$

---

🐦 **Qwen2.5-VL-32B-Instruct**

To solve this problem, we need to determine the force between objects $A$ and $B$ at the moment $B$ is placed on top of $A$. Let's break it down step by step.

. . .

4. **Force Analysis at the Moment $B$ is Placed:**
  - At the moment $B$ is placed, the spring has not yet had time to extend further to balance the new total weight. The spring force remains $60\,\text{N}$ (the initial force that balances the weight of $A$).
  - The net force on $A$ is the sum of the spring force and the force from $B$:$F_s + F_B = m_A g$.
  - Here, $F_s = 60\,\text{N}$ and $m_A g = 60\,\text{N}$, so: $60\,\text{N} + F_B = 60\,\text{N}$.
  - Solving for $F_B$, we get: $F_B = 0\,\text{N}$.
5. **Re-evaluate the Situation:**
  - The above analysis is not correct because we need to consider the instantaneous force between $A$ and $B$ due to the sudden addition of $B$.
  - The force between $A$ and $B$ is the force $B$ exerts on $A$, which is the weight of $B$: $F_B = m_B g = 4\,\text{kg} \cdot 10\,\text{m/s}^2 = 40\,\text{N}$.

$$\boxed{40}$$

---

🐳 **MM-EUREKA-32B**

To solve this problem, we need to determine the force between objects $A$ and $B$ at the moment $B$ is placed on top of $A$. Let's break the problem into steps:

. . .

4. **Applying Newton's Second Law to $A$**

$$F_{\text{net}} = F_B + F_A - F_{\text{spring}}$$
$$= F_B + 60 - 60 = F_B.$$

The net force on $A$ causes an upward acceleration $a$, which is the same as the acceleration of $B$ because they are in contact.

5. **Analyzing the Forces on $B$**
  - The forces on $B$ are:
    - The weight of $B$: $F_B = 40\,\text{N}$ (downward),
    - The force from $A$: $F_{\text{contact}}$ (upward).

6. **Applying Newton's Second Law to $B$**

$$F_B - F_{\text{contact}} = m_B a.$$

7. **Finding the Acceleration $a$**
  - The total mass of the system is $m_{\text{total}} = m_A + m_B = 6 + 4 = 10\,\text{kg}$.
  - The net force on the system is the extra weight from $B$: $F_{\text{net, total}} = 40\,\text{N}$.
  - Using Newton's Second Law for the entire system: $F_{\text{net, total}} = m_{\text{total}} a \Rightarrow 40 = 10a \Rightarrow a = 4\,\text{m/s}^2$.

8. **Finding the Contact Force $F_{\text{contact}}$**

$$F_B - F_{\text{contact}} = m_B a$$
$$40 - F_{\text{contact}} = 4 \cdot 4 = 16$$
$$F_{\text{contact}} = 40 - 16 = 24\,\text{N}.$$

<answer>The answer is $\boxed{C}$. </answer>

Figure 7: Example from physics. Red highlights Qwen2.5-VL-32B-Instruct's reasoning errors. Green shows MM-Eureka-32B's correct reasoning.

**Question:**
At room temperature, a 0.100 mol/L NaOH solution is used to titrate 25.00 mL 0.100 mol/L $H_2C_2O_4$ solution, and the resulting titration curve is shown in the figure (the volume of the mixed solution can be considered as the sum of the volumes of the solutions before mixing). Which of the following statements is incorrect? ( )

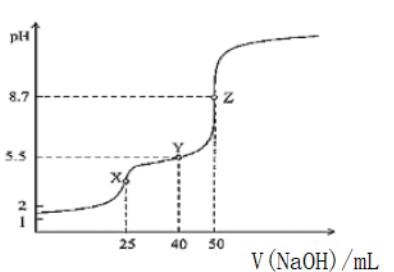

**A**. In the solution at point X: $c(\text{Na}^+) > c(\text{HC}_2\text{O}_4^-) > c(\text{C}_2\text{O}_4^{2-})$
**B**. Observing the solution at point Y indicates that the degree of ionization of $\text{HC}_2\text{O}_4^-$ is greater than the degree of hydrolysis of $\text{C}_2\text{O}_4^{2-}$
**C**. In the solution at point Z: $c(\text{Na}^+) > c(\text{C}_2\text{O}_4^{2-}) > c(\text{HC}_2\text{O}_4^-) > c(\text{OH}^-) > c(\text{H}^+)$
**D**. Throughout the titration process: $c(\text{Na}^+) + c(\text{H}_2\text{C}_2\text{O}_4) + c(\text{HC}_2\text{O}_4^-) + c(\text{C}_2\text{O}_4^{2-}) = 0.100 \text{mol/L}$

**Answer:** $C$

---

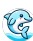 **Qwen2.5-VL-32B-Instruct**

**1. Understanding the Reaction:**
- The titration is between a strong base (0.100 M NaO) and a weak dicarboxylic acid (0.100 M $H_2CO_4$).

- The reaction steps are:

  1. $H_2CO_4 + NaO \rightarrow NaHC_2O_4 + H_2O$
  2. $NaHC_2O_4 + NaO \rightarrow Na_2CO_4 + H_2O$

**2. Key Points on the pH Curve:**
- **Point X (pH ≈ 2):** This is the initial point where only $H_2CO_4$ is present. The solution is acidic.

- **Point Y (pH ≈ 5.5):** This is the first equivalence point where all $H_2CO_4$ has been neutralized to $NaHC_2O_4$ (solutions of $NaHC_2O_4$ are acidic due to the protonation of the conjugate base).

- **Point Z (pH ≈ 8.7):** This is the second equivalence point where all $NaHC_2O_4$ has been neutralized to $Na_2CO_4$ (solutions of $Na_2CO_4$ are basic due to the deprotonation of the conjugate base).

...
**4. Final Answer:** The only option that is **incorrect** is Option $\boxed{A}$.

---

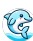 **MM-EUREKA-32B**

To solve this problem, we need to analyze the titration of a weak dicarboxylic acid $H_2C_2O_4$ (oxalate) with a strong base NaOH. The titration process involves multiple steps, and we must consider the equilibrium and ionic concentrations at different points on the titration curve. Let's break the problem into parts and solve it step by step.
**1. Understanding the titration process:**
$H_2C_2O_4$ is a weak dicarboxylic acid, which undergoes two steps of dissociation:

$$H_2C_2O_4 \rightleftharpoons H^+ + HC_2O_4^-$$
$$HC_2O_4^- \rightleftharpoons H^+ + C_2O_4^{2-}$$

The titration is carried out with NaOH, a strong base, which will neutralize the $H^+$ ions step by step.
**2. Key points on the titration curve:**
**Point X:** 25 mL of NaOH added.
- At this point, the first equivalence point is reached. All $H_2C_2O_4$ is converted to $NaHC_2O_4$ (sodium hydrogen oxalate).

- The solution contains $NaHC_2O_4$, and the pH is around 2, indicating that $HC_2O_4^-$ is a weak acid.
**Point Y:** 40 mL of NaOH added.
- This is between the first and second equivalence points. The solution contains a mixture of $NaHC_2O_4$ and $Na_2C_2O_4$.

- The pH is 5.5, indicating that the solution is slightly acidic due to the dissociation of $HC_2O_4^-$ being greater than the hydrolysis of $C_2O_4^{2-}$.
**Point Z:** 50 mL of NaOH added.
- At this point, the second equivalence point is reached. All $H_2C_2O_4$ is converted to $Na_2C_2O_4$ (sodium oxalate).

- The solution contains $Na_2C_2O_4$, and the pH is 8.7, indicating that $C_2O_4^{2-}$ is a strong base.

...
<answer>The answer is $\boxed{C}$. </answer>

---

Figure 8: Example from chemistry. Red highlights Qwen2.5-VL-32B-Instruct's reasoning errors. Green shows MM-Eureka-32B's correct reasoning.

**Question:**
Scientists isolated gene A from a certain organism and dissociated it into two single strands. One of these strands was hybridized with the messenger RNA of gene A, resulting in the pattern shown in the image. An unreasonable explanation for this is (  )
**A**. Positions 1, 3, 5, and 7 are connected through complementary base pairing
**B**. Heating to a certain temperature can separate the two strands
**C**. The diagram contains 8 types of nucleotides, 5 types of bases, and 2 types of base pairs
**D**. Since fragments 2, 2, and 6 cannot bind with the messenger RNA, fragments 2, 4, and 6 belong to non-gene segments of the DNA molecule

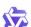

**Answer:** *D*

---

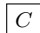 **Qwen2.5-VL-32B-Instruct**

. . .
**2. Analyzing Each Option:**
. . .
**Option C: The figure contains 8 types of nucleotides, 5 types of bases, and 2 types of base pairings**
- In the context of RNA and DNA, there are 4 types of nucleotides in each (A, U, G, C for RNA; A, T, G, C for DNA). When they anneal, the possible base pairings are A-T (or A-U), and G-C. This means there are 4 types of nucleotides per molecule, 5 types of bases (A, T, U, G, C), and 2 types of base pairings (A-T/U, G-C).
- The statement that there are 8 types of nucleotides is unreasonable because there are only 4 types of nucleotides in each (DNA and RNA).

**Option D: Since segments 2, 4, and 6 cannot bind with the messenger RNA, segments 2, 4, and 6 belong to non-gene segments of the DNA molecule**
- This is reasonable because segments 2, 4, and 6 do not anneal with the mRNA, which suggests that these segments do not code for the protein and are likely non-coding (non-gene) segments of the DNA.

$$\boxed{C}$$

---

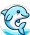 **MM-EUREKA-32B**

To solve this problem, we need to analyze the image and combine it with the principle of hybridization between DNA and messenger RNA (mRNA). Let's reason step by step:
. . .
**2. Analyzing the options**:
. . .
**C. The image contains 8 types of nucleotides, 5 types of bases, and 2 types of base pairings**:
- Nucleotides:
    - DNA contains 4 types of deoxyribonucleotides (dA, dT, dC, dG)
    - mRNA contains 4 types of ribonucleotides (A, U, C, G)
    - Total: 8 types of nucleotides.
- Bases:
    - DNA contains 4 bases (A, T, C, G)
    - mRNA contains 4 bases (A, U, C, G)
    - Since U replaces T in RNA, there are only 5 distinct bases (A, T/U, C, G).
- Base pairing:
    - In DNA-mRNA hybridization, the pairings are A–U, T–A, C–G, G–C.
    - However, only A–U and C–G actually form during hybridization.
    - Thus, only 2 types of base pairings exist.

**D. Since fragments 2, 4, and 6 do not hybridize with mRNA, they belong to non-genic segments of DNA**:
- The unhybridized regions (2, 4, 6) do not pair with mRNA, but this does not directly imply that they are non-genic segments.
- In genomic DNA, unpaired regions could be non-coding parts of a gene (e.g., introns, enhancers), or regulatory regions.
- Therefore, we cannot conclude that they are non-genic simply because they do not hybridize.

**3. Conclusion**:
- Option D is incorrect because we cannot determine from hybridization alone that these regions are non-genic.

<answer>The answer is $\boxed{D}$. </answer>

---

Figure 9: Example from biology. Red highlights Qwen2.5-VL-32B-Instruct's reasoning errors. Green shows MM-Eureka-32B's correct reasoning.

