# OpenReview forum: "MM-Eureka: Toward Stable Multimodal Reasoning via Rule-based Reinforcement Learning with Policy Drift Control"
_TMLR — Accepted by TMLR_

### Review · Reviewer_DDzB · 2026-03-10

**Summary Of Contributions:**

**Summary:** This paper analyzes catastrophic mid-training collapse in long-horizon multimodal reasoning. To address this issue, the authors propose Clipped Policy Gradient Optimization with Policy Drift (CPGD). This objective replaces PPO’s density ratio with a log-ratio formulation and explicitly incorporates a policy drift term into the loss. The paper also introduces a new multimodal reasoning dataset and an open-source training/evaluation pipeline. Empirically, the authors report that CPGD improves training stability and leads to better performance.

**Strength:**
- The paper contributes a new multimodal reasoning dataset to the community and releases an open-source pipeline, which improves reproducibility and lowers the barrier for follow-up work.
- The paper is generally well organized and easy to follow.

**Weakness:**
- The presentation contains several minor issues that could be addressed with small edits (see Requested Changes).
- The analysis of the training-collapse behavior in Section 4.3 is limited to a single dataset (MMK12). This makes it difficult to assess whether the identified collapse mode and the proposed mitigation generalize. In addition, the hyperparameter choices used in this analysis are not sufficiently justified or supported.
- In Section 5.3 (RL Algorithm Comparison), the paper claims that CPGD offers superior training stability, but the evidence presented focuses primarily on final performance. The paper would be much more convincing if it included training curves alongside baseline comparisons.

**Additional Comments:**

I have a few questions:
- Can Proposition 1 be extended to the case with \alpha > 0, i.e., the objective of CPGD?
- Could you provide more details about the answer verification process?
- How is CPGD compared to the other methods, like PPO, Reinforce, etc?
- Could the authors elaborate on the mechanism behind the training collapse phenomenon? And how does the long reasoning horizon or multimodal interaction contribute to instability during training?

**Audience:**

Yes

**Audience Explanation:**

This paper will be of interest to researchers working on topics related to foundation models and reasoning. The dataset proposed in the paper provides a good source and benchmark for multimodal reasoning works.

**Broader Impact Concerns:**

There are no broader impact concerns.

**Claims And Evidence:**

No

**Claims Explanation:**

Some of the paper’s claims are only partially supported. In particular, the assertion that CPGD mitigates training collapse is not fully convincing. The analysis in Section 4.3 examines collapse behavior only on the MMK12 dataset, and it provides limited evidence or justification for the chosen hyperparameters. Moreover, the paper does not include training-phase comparisons between CPGD and baseline methods (e.g., learning curves). Without these results, the claim that the proposed approach stabilizes training remains weak.

**Requested Changes:**

**Major:**
- In Section 4.3, the analysis of training collapse is conducted only on the MMK12 dataset. To strengthen the claim that the identified collapse behavior is general, the authors should provide additional evidence from other datasets, such as Mavis and MultiMath. It would also be helpful to include a clearer justification for the hyperparameter choices used in this analysis.
- The paper claims that CPGD improves training stability, but the experimental section does not include training comparisons with baseline methods. Providing training curvesfor CPGD and the baselines would help strengthen this claim.

**Minor:**
- Please define the abbreviations used in Table 1.
- In the appendix, the numbering of theoretical results appears inconsistent: Proposition 2 should be Proposition 1, and Theorem 2 should be Theorem 1.

---

> ### Author Response · Authors · 2026-03-24
> **Response**
>
> We thank the reviewer for the careful reading and constructive feedback. We address the main concerns below and have revised the paper accordingly. In response, we have strengthened the paper with: (i) new collapse experiments on **DeepVision** (Appendix E), (ii) clearer hyperparameter justification and sensitivity analysis (Appendix E, Figure 5a), (iii) more explicit training-curve evidence (Figure 3 and Appendix E), and (iv) fixes to the presentation issues (Table 1, Appendix A).
>
> ### 1. Collapse analysis beyond MMK12
>
> We agree that evidence on a second dataset is important. In the revised paper, we add a new collapse study on **DeepVision** [4](Appendix E, Figure 5b). We compare **REINFORCE++**, **GRPO**, and **CPGD** under matched settings. The same qualitative conclusion holds: **GRPO and REINFORCE++ still exhibit mid-training collapse, while CPGD remains stable**. This strengthens our claim that the failure mode is not specific to MMK12.
>
> ### 2. Hyperparameter justification
>
> We appreciate this comment and have clarified the hyperparameter rationale in the revision. Our choices were guided by established practice in recent concurrent works rather than selected arbitrarily: **temperature / top-p / top-k** follow common settings in recent open RL / reasoning works [1–3]; **learning rate = 1e-6** is consistent with Qwen-based RL training in [1,3]; and **batch size = 128** follows common OpenRLHF-style implementations [2,3]. We also performed empirical validation over batch sizes `{64, 128, 256}` and learning rates `{5e-7, 1e-6}`, and selected the setting that performed best overall across algorithms.
>
> We further ran additional sensitivity checks for **GRPO** (Appendix E, Figure 5a). Collapse still occurs across multiple settings: at around steps **800 / 500 / 400 / 300** for batch sizes **64 / 128 / 256 / 512**, and around **800 / 500** for learning rates **5e-7 / 1e-6** (with batch size 128). This suggests that the instability is not due to a single hyperparameter choice.
>
> ### 3. Training-phase evidence for stability
>
> We note that the original submission already included training-phase evidence in **Figure 3**, though we agree it could have been presented more prominently. Figure 3 directly visualizes the collapse of several baselines versus the stable trajectory of CPGD. In particular, while methods such as **GRPO**, **RLOO**, and **REINFORCE++** collapse around the middle of training, **CPGD maintains stable and monotonic improvement throughout**.
>
> ### 4. Minor presentation issues
>
> We thank the reviewer for catching these issues. In the revision, we define the abbreviations in **Table 1** (caption now includes definitions for Geo, Func, MCQ, FB, Img., QA, CoT) and fix the appendix numbering inconsistency (**Proposition 1 / Theorem 1**, Appendix A).
>
> ### 5. Additional questions
>
> **Q: Can Proposition 1 be extended to the case $\alpha > 0$?**
> Yes. The argument extends naturally: the additional policy-drift term introduces a pull-back effect toward $\pi_{\mathrm{old}}$, making the policy deviation smaller than in CPG ($\alpha=0$). We clarify this extension in the **revised appendix** (Appendix A.2, Proposition 2).
>
> **Q: Could you provide more details about answer verification?**
> Yes. We expand this part in the revision (Appendix C.3). The paper already uses **Math-Verify** for answer parsing/comparison and a separate **format reward** for checking the required `<think>...</think><answer>...</answer>` structure. We further clarify verification details in the **revised manuscript**.
>
> **Q: How is CPGD compared to PPO / REINFORCE-style methods?**
> Compared with **PPO/GRPO-style methods**, CPGD retains the proximal-update spirit but removes the importance-sampling ratio from the gradient-carrying term, using clipped **log-ratios** plus an explicit **forward-KL drift penalty**. Like GRPO, it also does **not require a critic network**. Compared with **REINFORCE / vanilla PG**, CPGD adds both a **clip mechanism** and **policy-drift control**, which are important for preventing collapse and reward-hacking behaviors.
>
> **Q: What is the collapse mechanism, and why is long-horizon multimodal reasoning more unstable?**
> Our explanation is that ratio-based objectives can amplify harmful updates, especially under negative advantages, and one-sided clipping does not fully prevent this. In long-horizon multimodal reasoning, sparse rewards and longer trajectories make such unstable updates more damaging, so a few problematic samples can dominate the gradient and trigger collapse.
>
>
> [1] Open-Reasoner-Zero: An Open Source Approach to Scaling Up Reinforcement Learning on the Base Model
>
> [2] OpenRLHF: An Easy-to-use, Scalable and High-performance RLHF Framework
>
> [3] LMM-R1: Empowering 3B LMMs with Strong Reasoning Abilities Through Two-Stage Rule-Based RL
>
> [4] DeepVision-103K: A Visually Diverse, Broad-Coverage, and Verifiable Mathematical Dataset for Multimodal Reasoning

---

### Review · Reviewer_UzQw · 2026-03-19

**Summary Of Contributions:**

**Summary**

The paper studies instability in rule-based reinforcement learning for long-horizon multimodal reasoning. The authors argue that PPO/GRPO-style ratio-based objectives can amplify harmful policy shifts under sparse multimodal rewards, which may trigger catastrophic mid-training collapse. To address this, they propose CPGD, which removes the importance ratio from the main gradient-carrying term, keeps proximal control via clipping in log-ratio form, and adds an explicit policy-drift regularizer. The paper also introduces MMK12, a multimodal K12 reasoning dataset with 15,616 training problems and 2,000 evaluation questions across mathematics, physics, chemistry, and biology. Empirically, the paper reports that CPGD yields more stable training and improves performance over GRPO, RLOO, and REINFORCE++ on several multimodal reasoning benchmarks.

**Strengths**

1. The paper addresses an important practical issue.

2. The empirical section is reasonably substantial.

3. The dataset contribution seems potentially useful. MMK12 could become a valuable benchmark for studying multimodal RL stability and cross-domain generalization if it is released and documented carefully.

**Weakness**

1. My main reservation is that the theoretical support feels more limited than the paper’s framing suggests.

2. The method bundles several modifications together: removing the ratio from the gradient-carrying term, introducing policy drift control, clipping in log-ratio space, and using a custom KL estimator. The experiments suggest the full recipe works, but the contribution of each ingredient is still not completely isolated.

**Audience:**

Yes

**Audience Explanation:**

Yes. Training stability in multimodal RL is a relevant topic, and I expect a meaningful part of the TMLR audience would care about both the failure mode identified here and the proposed stabilization approach.

**Claims And Evidence:**

Yes

**Claims Explanation:**

**Strengths**:  The diagnosis of training collapse in multimodal RL is interesting and, to me, the most compelling part of the paper. The stability plots and algorithm comparisons make the failure mode concrete rather than merely anecdotal. The authors compare against several natural RL baselines under shared settings and evaluate both on standard benchmarks and on MMK12. The reported gains are not tiny, and the stability evidence is fairly consistent with the paper’s main claim.

**Weaknesses**: The core theory is relatively stylized, and there is a noticeable gap between the clean update analyzed in the theory and the practical per-token objective actually optimized in implementation. From my perspective as someone with a theory background than in multimodal RL systems, this gap makes me hesitant to treat the theory as a strong justification of the full method rather than as intuition supporting the design; though I think it's completely fine, as the paper's contribution is mainly empirical.

**Requested Changes:**

1. Clarify more explicitly the gap between the theoretical update and the practical per-token implementation.

2. Provide cleaner isolated contributions of the main components of CPGD.

---

> ### Author Response · Authors · 2026-03-24
> **Response**
>
> We thank the reviewer for the thoughtful evaluation and for recognizing the practical importance of the stability issue and the dataset contribution. We address the two requested changes below.
>
> ### Q1: Theory-practice gap
>
> We thank the reviewer for this feedback and agree that a gap exists between the theoretical formulation and the practical implementation. We believe this gap does not diminish the value of the theory but reflects a principled evolution from theoretical insights to practical design. The theoretical formulation in Equation (1) provides the design framework for the practical algorithm. Each implementation detail is a reasonable improvement based on Equation (1), targeting more stable training. We have added a clarifying statement in Section 4.4 (Implementation) of the revised paper to make this explicit.
>
> - The response-level policy optimization term can be naturally decomposed into token-level form due to the natural additivity of logarithms. This decomposition enables integration with mainstream training frameworks such as OpenRLHF and veRL. It serves as a natural bridge between theory and practice.
>
> - The implementation of policy drift builds upon the gradient form of KL divergence and the additivity of logarithms. By introducing clipping at the gradient level, we ensure that when the policy ratio deviates, the gradient provides a corrective signal rather than exacerbating the deviation. This design directly stems from the theoretical concern for training stability.
>
> - The weighted advantage mechanism is orthogonal to the preceding theoretical analysis. It constitutes a set of plug-and-play practical techniques for dynamically adjusting gradient weights across samples. Our ablation studies confirm that this mechanism significantly improves training effectiveness, representing a key component of what makes the practical implementation work.
>
> In summary, the relationship between Equation (1) and Equation (2) exemplifies a **virtuous cycle of theory guiding practice and practice validating theory**, rather than a disconnect between the two.
>
>
> ### Q2: Isolated component contributions
>
> We thank the reviewer for this constructive suggestion. We have added a dedicated component ablation study in the revised paper (Appendix C.8, Tables 9--10) that clearly isolates the contribution of each component. Below, we summarize the results.
>
> **Table 9: Component Ablation** (Appendix C.8)
>
> | Model | MathVista | MathVerse | MathVision | Olypamid | WeMath | MMK12 | Overall |
> |-------|-----------|-----------|------------|----------|--------|-------|---------|
> | **CPGD (STD weight)** | **74.0** | **50.6** | **28.3** | **21.4** | **68.3** | **65.3** | **1.11** |
> | PG | 67.8 | 42.0 | 22.5 | 8.0 | 58.6 | 65.9 | 0.89 |
> | PGD | 64.2 | 41.1 | 20.8 | 7.5 | 58.3 | 67.3 | 0.86 |
> | CPG | 72.7 | 52.3 | 27.6 | 20.8 | 70.7 | 66.2 | 1.11 |
>
> **Table 10: Weighting Factor Ablation** (Appendix C.8)
>
> | Model | MathVista | MathVerse | MathVision | Olypamid | WeMath | MMK12 | Overall |
> |-------|-----------|-----------|------------|----------|--------|-------|---------|
> | unprocessed rewards | 69.1 | 40.2 | 21.8 | 3.5 | 59.7 | 67.2 | 0.85 |
> | equal weight | 73.1 | 51.1 | 27.2 | 20.8 | 67.9 | 65.8 | 1.09 |
> | clip-filter-like weight | 73.4 | 51.4 | 25.9 | 21.5 | 70.2 | 67.3 | 1.10 |
> | STD weight | 74.0 | 50.6 | 28.3 | 21.4 | 68.3 | 65.3 | 1.11 |
>
> The results reveal a clear hierarchy of component importance:
>
> **1. Clip Mechanism (Most Critical)**: The clip mechanism is the most critical component. Comparing CPG vs. PG: removing clipping causes a dramatic performance drop from 1.11 to 0.89 overall. This confirms our theoretical analysis—clipping prevents response length collapse and ensures proximal policy updates, which is essential for stable training.
>
> **2. Weighted Advantage (Effective Enhancement)**: The weighted advantage mechanism significantly improves performance. Using raw unprocessed rewards yields only 0.85 overall, while proper advantage weighting (STD weight or clip-filter-like) achieves 1.10-1.11. This demonstrates that advantage normalization is crucial for effective learning.
>
> **3. Policy Drift (Secondary Stabilization)**: Policy drift provides secondary benefits. Comparing PGD vs. PG shows minimal improvement (0.86 vs. 0.89), and adding it to CPG yields no additional gain (CPGD vs. CPG: both 1.11). Its primary role is as a safeguard against extreme ratio deviations rather than a primary performance driver.

---

### Review · Reviewer_JzRi · 2026-03-22

**Summary Of Contributions:**

In this paper, the authors study the problem of stable and performant multimodal reinforcement learning. To this end, they introduce CPGD, an interesting RL objective that prevents training collapse by eliminating ratio-induced amplification (i.e., with log ratio-based clipping) and utilizing explicit policy drift constraints (k3 based regularization). In addition, to support reliable training and diagnose instability, they developed MMK12, a comprehensive K12-level multimodal dataset w/ over 15k training problems and 2k evaluation examples. By applying CPGD to the MMK12 dataset, they successfully trained the MM-Eureka models, showing stable, long-horizon training without catastrophic mid-training collapse and achieving performance improvements.

Pros:

1. The proposed CPGD objective has an interesting design and includes mathematical proofs to demonstrate that it can prevent ratio-related policy shifts while preserving guarantees for policy improvement.

2. To train and eval the proposed method, the authors built a large-scale, high-quality dataset with over 15k training examples and 2k evaluation examples across math, physics, chemistry and biology etc.

3. The resulting MM-Eureka model can improve the base models' performance on a variety of tasks, matching or exceeding the performance of larger models and achieving competitive results against in some tasks.

Cons:

1. The proposed CPGD objective relies on incremental mathematical adjustments to existing policy gradient frameworks, rather than introducing a fundamentally novel reinforcement learning paradigm / algorithm. In addition, replacing the importance-sampling ratio with a log ratio removes off-policy correction from the objective, which may be problematic for large-scale / async RL training.

2. While being competitive among open-source models, MM-Eureka models still trails behind top-tier closed-source models on the challenging benchmarks such as MathVision. Also, the choice of base models and baselines is slightly outdated. To prove that the proposed CPGD framework is robust, the authors should include experiments utilizing more recent open-source multimodal models.

**Audience:**

Yes

**Audience Explanation:**

This submission introduces an effective RL algorithm for multimodal post-training, making it a valuable and highly relevant contribution to the broader ML community.

**Claims And Evidence:**

Yes

**Claims Explanation:**

For the most part, the claims made in the submission are supported by accurate, convincing, and clear evidence. The authors back up their statements with theoretical proofs, empirical testing and qualitative results.

**Requested Changes:**

Conduct additional experiments using more recent multimodal foundation models, such as Qwen3-VL / Molmo2 (smaller variants should suffice), to demonstrate the method's effectiveness on current state-of-the-art architectures.

---

> ### Author Response · Authors · 2026-04-02
>
> We thank the reviewer for the comment. We agree that CPGD is developed within the policy gradient framework, but we do not view this as a limitation. Our contribution is to identify a previously overlooked instability issue in existing LLM RL methods and to introduce a targeted, theoretically motivated fix. In this sense, CPGD is not an arbitrary modification, but a principled improvement driven by problem diagnosis. More importantly, recent LLM RL methods such as GRPO, REINFORCE++, and RLOO are also developed through incremental refinements of policy gradient methods rather than entirely new RL paradigms. We therefore believe that identifying a critical flaw and resolving it in a simple, effective, and empirically validated way is a meaningful contribution.
>
> We also thank the reviewer for raising the point about the importance-sampling ratio. While its role is well motivated in general RL, our empirical results suggest that it may be less beneficial in LLM RL. A key reason is that current LLM RL pipelines typically fix the number of gradient updates per rollout generation round (i.e., one PPO epoch), keeping the policy close to the sampling distribution throughout training. In this approximately on-policy regime, the importance-sampling correction provides limited benefit. Meanwhile, token-level ratios compound across autoregressive generation and introduce substantial variance, which can destabilize training. Consistent with this analysis, across 7B–32B models, removing the importance-sampling ratio consistently improves performance over objectives that retain it, such as GRPO and REINFORCE++.
>
> Regarding the concern about the choice of base models, we respectfully note that our experiments already cover a broad range of model families, scales, and modalities. Specifically, we have evaluated CPGD on: (1) InternVL2.5-8B, a different model family from Qwen; (2) Qwen2.5-VL-7B, our primary backbone; (3) Qwen2.5-VL-32B, a significantly larger scale; and (4) Qwen3-8B (text-only), extending CPGD beyond the multimodal domain to pure LLM RL. These results are presented in Appendix C.6 (Tables 6, 7, 8). Following the reviewer's suggestion, we have further conducted experiments on Qwen3-VL-8B-Instruct, a recent state-of-the-art multimodal model. As shown in the newly added Table 7 (Appendix C.6), training Qwen3-VL-8B-Instruct with MMK12 and CPGD yields consistent improvements across all six benchmarks, confirming that our data-algorithm combination generalizes effectively to newer model architectures. Taken together, these experiments span 3 model families (InternVL, Qwen2.5, Qwen3), 3 model scales (8B, 32B, 72B-competitive), and both multimodal and text-only settings, which we believe sufficiently demonstrates the robustness and generality of the overall framework.

---

### Decision · Action_Editor_CaJc · 2026-04-27

**Recommendation:** Accept as is

**Audience:**

Yes

**Audience Explanation:**

This paper is dealing with post-training RL collapse in multimodal models, which is of interest for a large swath of TMLR's audience.

**Claims And Evidence:**

Yes

**Claims Explanation:**

While there were some concerns about the authors overclaiming their results, the rebuttal provided by the authors addressed most (if not all) of the concerns of the reviewers.